

**The influence of episodic flooding on pelagic ecosystem in the East China Sea**
Chung-Chi Chen[1, *], Gwo-Ching Gong[2], Wen-Chen Chou[2], Chih-Ching Chung[2], Chih-Hao
Hsieh[3], Fuh-Kwo Shiah[2, 4], Kuo-Ping Chiang[2]
[1]Department of Life Science
National Taiwan Normal University
No. 88, Sec. 4, Ting-Chou Rd., Taipei 116, Taiwan
[2]Institute of Marine Environmental Chemistry and Ecology
National Taiwan Ocean University
Keelung 202-24, Taiwan
[3]Institute of Oceanography
National Taiwan University
Taipei 10617, Taiwan
[4]Research Center for Environment Changes
Academia Sinica, NanKang
Taipei 115, Taiwan
*: Corresponding author:
Telephone: 886.2.2930.2275
Fax #: 886.2.2931.2904
E-mail: ccchen@ntnu.edu.tw
Running header: Effect of flooding in the East China Sea



32                                              ABSTRACT

This study was designed to determine the effects of flooding on pelagic ecosystem in the East
China Sea (ECS), especially on plankton community respiration (CR). In July 2010, a flood
occurred in the Changjiang River. As a comparison, a variety of both abiotic and biotic
parameters were monitored, as well as in July 2009, a non-flooding period. During the flooding,
the Changjiang diluted water (CDW) zone covered almost two thirds of the ECS, which was
approximately six times that of the non-flooding period. The mean nitrate concentration was
higher in 2010 (6.2 μM) than in 2009 (2.0 μM). However, during the 2010 flood, the mean values
of Chl $a$ and bacterial biomass were only slightly higher or even lower than in 2009. However,
the CR was still higher in 2010 than in 2009, with mean values of 105.6 and 73.2 mg C m$^{-3}$ d$^{-1}$,
respectively. The higher CR in 2010 could be attributed to vigorous plankton metabolic activities,
especially phytoplankton, at stations in the CDW zone, which were not characterized by low SSS
in 2009. In addition, zooplankton might be another important component contributing to the high
CR rate observed in 2010. Furthermore, there was a significant amount of $f$CO$_2$ drawdown in the
2010 flood. These results suggest that the flood in 2010 had a significant effect on the carbon
balance in the ECS. This effect might become more pronounced in the future, as extreme rainfall
events and flooding magnitudes are predicted to increase globally due to climate change.



Keywords: Bacteria; Dissolved inorganic nutrients; East China Sea; Phytoplankton; Plankton

community respiration; Primary production; Yangtze River



 INTRODUCTION

Riverine run-off has a profound effect on the production of organic carbon and its
consumption in coastal ecosystems (e.g., Dagg et al., 2004; Hedges et al., 1997 and references
therein). Accompanying freshwater discharge, a substantial amount of dissolved inorganic
nutrients is routinely delivered into coastal regions, thus enhancing primary productivity (PP;
e.g., Dagg et al., 2004; Nixon et al., 1996). In addition, a large quantity of particulate and
dissolved organic matter is discharged via riverine input (e.g., Wang et al., 2012). High rates of
microbial metabolism associated with this discharge have been observed in marine environments
at local scales (e.g., Hedges et al., 1994; Malone and Ducklow, 1990). River plumes can extend
for hundreds of kilometers along the continental shelf, as in the case of the Amazon River (e.g.,
Müller-Karger et al., 1988). Overall, the effects of river plumes on coastal ecosystems are
strongly related to the volume of freshwater discharge (e.g., Chen et al., 2009; Dagg et al., 2004;
Tian et al., 1993). Thus, understanding how freshwater discharge influences coastal ecological
processes is an important factor in exploring global carbon cycling in the adjacent seas. Under the
current conditions of climate change, such heavy freshwater discharge events are predicted to
become even more pronounced in the near future because of the dramatic increases in extreme
rainfall events and floods predicted to occur throughout the world (Christensen and Christensen,
2003; Knox, 1993; Milly et al., 2002; Palmer and Ralsanen, 2002).



The East China Sea (ECS) has an approximate area of $0.5 \times 10^6$ km$^2$ and is the largest

marginal sea in the western Pacific. A tremendous amount of freshwater (956 km$^3$ yr$^{-1}$) is

discharged annually into the ECS, notably by the Changjiang (a.k.a Yangtze) River, which is the

fifth largest river in the world in terms of volume discharge (Liu et al., 2010). On average, the

maximum amount of discharge occurs in July, and mean monthly values have ranged from

33,955 to 40,943 m$^3$ s$^{-1}$ in years of normal weather during the past decade (Gong et al., 2011; Xu

and Milliman, 2009). After having been discharged into the ECS, freshwater mixes with seawater

to form the Changjiang diluted water (CDW) zone, the salinity (SSS) of which is $\leq 31$ psu (e.g.,

Beardsley et al., 1985; Gong et al., 1996). In the CDW, especially in summer, the regional carbon

balance is regulated by high rates of plankton community respiration (CR) and PP (Chen et al.,

2006; Gong et al., 2003). The rates of CR were also positively associated with the riverine flow

rates (Chen et al., 2009). However, few previous studies have shown the effects of floods on

biological activity in the ECS (Chung et al., 2014; Gong et al., 2011). Historically, the threshold

discharge rates during Changjiang River flooding periods have been estimated to be $4\text{-}6 \times 10^4$ m$^3$

s$^{-1}$ (Committee, 2001). However, in recent decades, the frequency and magnitude of the

Changjiang River flooding events have increased, and this has been attributed to extreme

monsoon rainfall associated with climate warming (Jiang et al., 2007; Yu et al., 2009);

collectively, these observations suggest that the influence of flooding on the ECS shelf ecosystem



has intensified. Therefore, it is worthwhile exploring the responses of the biological activities and
ecological processes in the ECS to the periodic flooding of the Changjiang River.

In July 2010, a large flood occurred in the Changjiang River (Gong et al., 2011). This event

provided an opportunity to understand how flooding affects the ECS shelf ecosystem.
Comparative analyses were conducted to examine a number of variables, including physical,
chemical, and biological parameters, during a period (July 2009) when the riverine flow was
relatively low. The main objective of this study was to reveal the effects of the riverine input of
dissolved inorganic nutrients on the plankton communities that support heterotrophic processes in
the ECS shelf ecosystem between periods of non-flooding and flooding. To evaluate the
differences between these periods, variations in biological variables were compared with CR in
order to elucidate their relative importance to CR. In addition, the relationship between CR and
the fugacity of $CO_2$ ($fCO_2$) was examined to determine the contribution of the plankton
communities to variations in $fCO_2$ in periods of non-flooding and flooding.

MATERIALS AND METHODS

**Study area and sampling.** This study is part of the Long-term Observation and Research of

the East China Sea (LORECS) program. Samples were collected from the ECS in the summers of
2009 (June 29 to July 13) and 2010 (July 6 to 18) during two cruises on the *R/V Ocean*
*Researcher I*. The sample stations were located throughout the ECS shelf (Fig. 1). In July 2010,



the mean monthly discharge from the Changjiang River reached 60,527 $m^3$ $s^{-1}$, which was
significantly higher than the monthly discharge (33,955 $m^3$ $s^{-1}$) in the non-flooding year of 2009
(Gong et al., 2011; Yu et al., 2009). Water samples were collected using Teflon-coated Go-Flo
bottles (20 L, General Oceanics Inc., USA) mounted on a General Oceanic Rosette® assembly
(Model 1015, General Oceanics Inc., USA). At each station, six to nine samples were taken at
depths of 3 to 50 m, depending on the depth of the water column. Sub-samples were taken for
immediate analyses (dissolved inorganic nutrients, chlorophyll *a* [(Chl *a*)], and bacterial
abundance) and on-board incubation of PP and plankton CR. The methods used to collect the
hydrographic data and analyze the aforementioned response variables followed Chen et al. (2006;
2013; 2009). Descriptions of the methods used are presented briefly in the following sections. It
should also be noted that portions of these results were published by Chung et al. (2014) and
Gong et al. (2011).

**Physical and chemical hydrographics.** Temperature, salinity, and transparency were

recorded throughout the water column using a SeaBird CTD (USA). Photosynthetically active
radiation (PAR) was measured throughout the water column using an irradiance sensor ($4\pi$; QSP-
200L). The depth of the euphotic zone ($Z_E$) was taken as the penetration depth of 1% of surface
light. The mixed layer depth ($M_D$) was based on the potential density criterion of 0.125 units
(Levitus, 1982).



A custom-made flow-injection analyzer was used for dissolved inorganic nutrient (e.g.,
nitrate, phosphate, and silicate) analysis (Gong et al., 2003). Integrated values for the nitrates and
other variables in the water column above the $Z_E$ were estimated using the trapezoidal method, in
which depth-weighted means are computed from vertical profiles and then multiplied by $Z_E$ (e.g.,
Smith and Kemp, 1995). The average nitrate concentration over $Z_E$ was estimated from the
vertically integrated value divided by $Z_E$. This calculation was adopted to determine the values of
the other variables.
The fugacity of $CO_2$ ($f CO_2$) in the surface waters was calculated from dissolved inorganic
carbon (DIC) and total alkalinity (TA) data using a program designed by Lewis and Wallace
(1998). For details of the TA and DIC measurements, please see Chou et al. (2007).
**Biological variables.** The water samples taken for Chl *a* analysis were immediately filtered
through GF/F filter paper (Whatman, 47 mm) and stored in liquid nitrogen. The Chl *a* retained on
the GF/F filters was quantified fluorometrically (Turner Design 10-AU-005; Parsons et al., 1984).
When applicable, Chl *a* was converted to carbon units using C:Chl values of 52.9, estimated from
shelf waters of the ECS (Chang et al., 2003). To estimate total content of Chl *a* over $Z_E$ integrated
for the ECS and the CDW (please see below for details.), Surfer 11 (Golden Software, Inc.) was
used. This estimation was also adopted to determine the values for heterotrophic bacteria and
zooplankton.





143  Heterotrophic bacteria samples were fixed in paraformaldehyde at a final concentration of

144 0.2% (w/v) in the dark for 15 min. They were then immediately frozen in liquid nitrogen and kept

145 at -80°C prior to analysis. The heterotrophic bacteria were stained with the nucleic acid-specific

146 dye SYBR® Green I (emission = $530 \pm 30$ nm) at a final concentration of $10^{-4}$ dilution of a

147 commercial solution (Molecular Probes Inc., Oregon, USA) (Liu et al., 2002). They were then

148 identified and enumerated using a flow cytometeter (FACSAria, Becton-Dickinson Co., New

149 Jersey, USA). Known numbers of fluorescent beads (TruCOUNT Tubes, Becton-Dickinson)

150 were simultaneously used to calculate the original cell abundance in each sample. Bacterial

151 abundance was converted to carbon units using a conversion factor of $20 \times 10^{-15}$ g C cell$^{-1}$

152 (Hobbie et al., 1977; Lee and Fuhrman, 1987).

153  Zooplankton samples were collected across the whole water column at selected stations

154 using a 330-μm mesh net with a 160-cm diameter opening. Zooplankton samples were digitized

155 to extract the size information (i.e., body width and length) using the ZooScan integrated system,

156 and the size information was used to calculate the ellipsoidal bio-volume of zooplankton (Garcia-

157 Comas, 2010). The biomass (carbon units) of zooplankton was then calculated using the

158 estimated bio-volume following equations of Alcaraz et al. (2003). To estimate the biomass over

159 $Z_E$, the total biomass of zooplankton over the whole water column was multiple by the fraction of

160 "$Z_E$ relative to depth of the water column" at all stations.





Primary production (PP) was measured by the $^{14}$C assimilation method. The samples were
collected and incubated from three depths within the $Z_E$ at stations surveyed during the daylight
hours (Gong et al., 2003; Parsons et al., 1984). The samples were pre-screened through a 200-μm
woven mesh (Spectrum) and inoculated with $H^{14}CO_3^-$ (final conc. 10 μCi ml$^{-1}$) in clean 250-ml
polycarbonate bottles (Nalgene). The samples were incubated onboard for 2 hrs in chambers
filled with running surface seawater and illuminated by halogen bulbs with a light intensity
corresponding to the *in situ* irradiance levels (Gong et al., 1999). Following each incubation, the
samples were filtered on GF/F filters (Whatman, 25 mm), acidified with 0.5 ml 2 N HCl, and
then left overnight. After immersion in 10 ml of a scintillation cocktail (Ultima Gold, Packard),
the total activity on the filter was counted using a liquid scintillation counter (Packard 1600).
Please note that PP was measured only at selected stations in 2010, but not in 2009.
The plankton community respiration (CR) was measured as the decrease in dissolved
oxygen ($O_2$) during dark incubation (Gaarder and Grann, 1927). CR was measured in samples
collected from most stations, with two initial and two dark treatment samples taken from 4-6
depths (depth intervals of 3, 5, 10, 15, 20, and/or 25 m depending on the depth of the water
column) within the $Z_E$ at each station. The treatment samples were siphoned into 350-mL
biological oxygen demand (BOD) bottles and incubated for 24 hrs in a dark chamber filled with
running surface water. Maximum temperature changes were 1.33 ± 0.81 and 2.70 ± 1.43℃ (mean



± SD) during each incubation in 2009 and 2010, respectively. The concentration of $O_2$ was
measured by a direct spectrophotometry method (Pai et al., 1993). The precision of this method
was calculated as the root-mean square of the difference between the duplicate samples and was
found to be 0.02 and 0.03 mg $L^{-1}$ in 2009 and 2010, respectively. The precision for initial samples
in both periods was < 0.01 mg $L^{-1}$. The difference in $O_2$ concentration between the initial and the
dark treatment was used to compute the CR. A respiration quotient of 1 was assumed in order to
convert the respiration from oxygen units to carbon units (Hopkinson Jr., 1985; Parsons et al.,

1984).

RESULTS and DISCUSSION

**Comparison of hydrographic patterns between flooding and non-flooding periods**
In July 2010, the Changjiang River flooded to a devastating extent, and this flood started in
late May or early June. The mean monthly water discharge was 60,527 $m^3$ $s^{-1}$, and the threshold
discharge rate was 4-6 x $10^4$ $m^3$ $s^{-1}$, making it the largest recorded flooding of the Changjiang
River over the last decade (http://yu-zhu.vicp.net/). This rate was almost two times larger than
that recorded in the non-flooding period in July 2009 (33,955 $m^3$ $s^{-1}$) (Gong et al., 2011; Yu et al.,
2009). During the flood, a tremendous amount of freshwater was delivered into the ECS, and the
low salinity of the sea surface (SSS ≤ 31 psu) covered almost two thirds of the continental shelf
(Fig. 1b). The SSS in the ECS during the 2010 flood was significantly lower than that during the

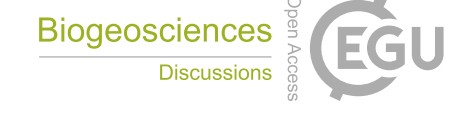

2009 non-flooding survey period; the mean (± SD for this and all parameters discussed
henceforth) values were 30.32 (± 3.60) and 32.62 (± 2.07) psu, respectively (Table 1). During
periods of high discharge from the river, particularly during the summer, the CDW zone is
generally distributed within the 60-m isobath region between the latitudes of 27 and 32 °N along
the coast (e.g., Beardsley et al., 1985; Gong et al., 1996). During the 2010 flood, the CDW
dispersed towards the east and south and reached as far as the 100-m isobath (Fig. 1b). The
substantial quantity of freshwater discharged into the ECS could also be seen in the coverage area
of the CDW (e.g., Gong et al., 2011); in the 2010 flood, the CDW area (111.7 x $10^3$ km$^2$),
approximately six times larger than in the 2009 non-flooding period (19.0 x $10^3$ km$^2$).

Although the mean SSS differed significantly between the flooding and non-flooding

periods, there was no difference in the temperature of the sea surface (SST; Table 1). The mean
values of SST in 2009 (26.8 ± 1.7) and 2010 (and 26.1 ± 2.2 ℃) were within the range of the
mean SST of the ECS in summer (Chen et al., 2009). The mixed layer depth ($M_D$) did not
significantly vary between survey periods: 13.7 (± 7.3) m in 2009 and 11.3 (± 6.6) m in 2010
(Table 1). However, the average $M_D$ was shallower than previously documented values, which
ranged from 16.8 to 28.2 m in the ECS during summer (Chen et al., 2009). Even though the mean
values of the euphotic depth ($Z_E$) were slightly deeper in 2009 (38.9±36.4 m) than in 2010
(33.4±17.3 m), there was no statistically significant difference between these depths (Table 1).



Regarding the $M_D$, the average $Z_E$ in the ECS was also shallower than in a previous study
conducted during the summer (Chen et al., 2009). The shallower $Z_E$ could also been indirectly
influenced by the transparency of the seawater. The average transparency in summer in the ECS
over the past six years (2003-2008) was 81.9% (C.C. Chen, unpublished data). The average
transparency values of the ECS in 2009 and 2010 were 76.7% and 80.5%, respectively (Table 1).
The averaged values for the CDW zone were relatively low in 2009 (70.0%) and relatively high
in 2010 (78.4%) compared to that of the trailing 6-year average (72.7%; C.C. Chen, unpublished
data). This might also explain why $Z_E$ in the CDW in 2009 was only 16.8 m (Table 1).

These findings suggest that the growth of phytoplankton might be limited by the availability

of light, especially in the CDW zone in 2009. Generally, the transparency of the coastal ocean
might be low during flooding periods due to riverine discharge of terrestrial matter. A low
transparency value was documented in June 2003 in the ECS, during which the CDW area was
43.1 x$10^3$ km$^2$ (Chen et al., 2009), and the average values for the ECS and the CDW were 70.9%
and 66.0%, respectively (C.C. Chen, unpublished data). Surprisingly, as stated above, the average
transparency in the ECS during the 2010 flood was similar to the value observed over the past six
years, and its value (78.4%) in the CDW in 2010 was even higher than that of the trailing 6-year
average (72.7%). This could be partially explained by the fact that most large particulates from
terrestrial sources might have been confined to and precipitated in the coastal region, not in the



expanded CDW region (e.g., Chung et al., 2012). Furthermore, it should also be noted that the
sampling period of 2010, even at the peak of the flood, was almost one month after the beginning
of this flood. Therefore, it is reasonable to speculate that plankton communities were in the late
phase of succession in this flood event (please refer to discussion in next section for more details
on this matter). The transparency during the 2010 sampling period might, then, have increased
due to organic matter (particulate and dissolved) having been uptaken and transferred to higher
trophic levels.

In general, an immense quantity of dissolved inorganic nutrients is delivered from the

Chinese coast into the ECS during the wet season, from May to September (Chen et al., 2013;
Chen et al., 2009; Gong et al., 1996). This study found a higher concentration of nitrates in the
ECS during flooding periods, mostly in the fluvial discharge of the Changjiang River. This
finding was supported by the negative linear relationship between SSS and nitrate concentration
in the ECS in 2010 ($r^2 = 0.37$, $p < 0.001$, n = 37). During the study period, there was also a
negative linear relationship between SSS and silicate concentration ($r^2 = 0.60$, $p < 0.001$, n = 37),
but not between SSS and phosphate concentration (data not shown). The comparison of the two
periods showed that the nitrate concentration in the surface water of the ECS was significantly
higher in the 2010 flood than in the 2009 non-flooding period, with mean values of 6.2 (± 9.8)
and 2.0 (± 5.3) μM, respectively (Table 1). This finding also applied to the average nitrate values





over $Z_E$ between both periods (data not shown). During the 2010 flood, the mean nitrate
concentration, either in the surface water or averaged over $Z_E$, was higher or comparable to that
documented during periods of high riverine discharge in the ECS (Chen et al., 2009; Gong et al.,
1996). Surprisingly, in the 2010 flood, nitrate levels reached 37.6 μM in the surface water, and
the highest nitrate concentrations were observed within the CDW (Fig. 1d).

As for phosphate concentration, there was no significant difference in values of the surface

water observed between the 2009 non-flooding period and the 2010 flood, with mean values of
0.13 (± 0.17) and 0.17 (± 0.30) μM, respectively (Table 1). Although mean phosphate values of
the surface water in the CDW zone were slightly higher in 2010 (0.23 μM) than in 2009 (0.13
μM), this difference was not statistically significant (Table 1). However, it should be noted that
there was one station with extremely high phosphate concentration (1.71 μM) in the surface water
in the CDW zone during the 2010 flood (Fig. 1f). Even so, in this period, the mean molar ratio of
nitrate to phosphate (N/P) was 22.3 ± 20.9. The high N/P molar ratio was even more pronounced
in the CDW, where it was higher than 16 at 14 of the 20 stations, with a mean value of 40.4 (±
22.6). This value was comparable to that of the CDW during high riverine flow periods in the
ECS in summer (Chen et al., 2006). During the non-flooding period, the N/P molar ratio was
lower than 16, with a mean value of 11.5 (± 20.8).

It has been suggested that phytoplankton growth might be regulated by the availability



and/or the N/P ratio of nutrients in the ECS (Gong et al., 1996; Harrison et al., 1990). The results
of this study indicate that in the 2009 non-flooding period, phytoplankton biomass might have
been regulated by the availability of dissolved inorganic nitrogen to a greater extent than it was
during the 2010 flood. However, in the 2010 flood, phytoplankton growth was likely limited by
phosphates. Phytoplankton growth limited by different inorganic nutrients in varying periods has
been observed in estuaries and coastal regions, such as Chesapeake Bay in the United States
(Fisher et al., 1992; Harding, 1994). In the ECS, phosphates have been frequently found as a
factor limiting phytoplankton growth, especially in the CDW (Chen et al., 2004; Gong et al.,
1996; Harrison et al., 1990).

**Plankton activities associated with the Changjiang River flood**

Following the discharge of fluvial nutrients into the ECS, phytoplankton are generally

abundant in the CDW region. The Chl $a$ concentration in the CDW even reached bloom criteria
(> 20 mg Chl m$^{-3}$) in past years in the ECS (Chen et al., 2009; Chen et al., 2003). Surprisingly,
the phytoplankton biomass was not as high as expected in this study, even though a high nitrate
concentration was observed during the 2010 flood. The mean values of Chl $a$ in the surface water
of the ECS in 2009 and 2010 were 0.98 (± 1.52) and 1.26 (± 1.27) mg Chl m$^{-3}$, respectively
(Table 1). However, these mean values were still at the high end of the Chl $a$ concentration range
normally documented in the ECS in the mid-summer through July/August period (Chen et al.,



2009). In both periods, the phytoplankton biomass in the surface water was generally higher in
the CDW than in other regions of the ECS (Fig. 1g and h). For example, in the 2010 flood, the
maximum Chl $a$ value reached 5.32 mg Chl m$^{-3}$ in the CDW (Table 1; Fig. 1h). In the 2010 flood,
the Chl $a$ values were positively related to nitrate and silicate concentrations (all $p < 0.001$), but
not phosphate concentrations ($p = 0.09$), in the surface water. The linear relationship between Chl
$a$ and phosphate values in the surface water, however, became significance ($p < 0.001$) if one
outlier with a markedly high phosphate concentration (1.71 μM) was excluded from this analysis
(Fig. 1f). In the 2009 non-flooding period, the Chl $a$ concentration was significantly, positively,
and linearly associated with concentrations of all measured nutrients: nitrate, silicate, and
phosphate ($p < 0.01$ in all cases).

The spatial distribution pattern of Chl $a$ documented in this study was similar to that found

in previous studies of the ECS (Gao and Song, 2005; Gong et al., 2011). However, it should be
noted that the CDW zone was extensive during the 2010 flood. Nevertheless, the phytoplankton
biomass in the surface water (Table 1), or averaged over $Z_E$ (data not shown), did not differ
significantly between 2009 and 2010. An effect of flooding on phytoplankton biomass was,
however, observed if using total content of Chl $a$ over $Z_E$ integrated for the entire ECS or the
CDW zone. The total Chl $a$ content in the ECS was higher in 2010 than in 2009, with values of
5.5 and 4.4 x 10$^6$ kg Chl $a$, respectively (Table 2). The Chl $a$ content in the CDW zone was even



higher in 2010 than in 2009, with values of 3.9 and 1.2 x $10^6$ kg Chl *a*, respectively (Table 2). In
the 2010 flood, PP in the surface water was high, with a mean (± SD) value of 62.1 (±33.8) mg C
$m^{-3}$ $d^{-1}$ (Table 1). This value was comparable to the high PP values documented in the ECS in
summer in prior works (Chen et al., 2009). In contrast, the PP:Chl *a* value was higher in the 2010
flood compared to that documented by Chen et al. (2009), with mean values of 27.1 (±17.2) and
19.7 (±5.5) mg C mg $Chl^{-1}$ $d^{-1}$, respectively.
Gong et al. (2011) estimated that over the past decade, the average rate of carbon fixation
during flooding periods was about three times higher than during non-flooding periods. During
the 2010 flood, the rate reached 176.0 x $10^3$ tons C $d^{-1}$ in the CDW (Gong et al., 2011). Gong et
al. (2011) also showed that the abundance of phytoplankton was twice as high in the CDW than
in the other regions. In the 2010 flood, the phytoplankton communities predominantly consisted
of diatoms, especially *Chaetoceros* spp., *Rhizosolenia* spp., and *Nitzschia* spp. (Gong et al.,
2011). However, the phytoplankton assemblage was not measured in 2009. In July 2007, when
the amount of freshwater discharge was similar to that of 2009 (Gong et al., 2011), the
phytoplankton in the CDW were predominantly diatoms and other algal taxa, including
dinoflagellates, coccolithophorids, and green algae (Chien, 2009). Picocyanobacteria, particularly
the phycocyanin-rich *Synechococcus,* were predominant in the CDW, and they showed similar
spatial distribution patterns in both 2009 and 2010 (Chung et al., 2014). In addition, the



phycoerythrin-rich *Synechococcus* covered most of the ECS continental shelf, but they were less
abundant in 2010 than in 2009 (Chung et al., 2014). Furthermore, the decreased presence of
*Prochlorococcus* was also observed in regions other than the CDW in both periods (Chung et al.,
2014). These results imply that the phytoplankton community assemblage might have differed
between the flooding and non-flooding periods investigated in this study, even though the
phytoplankton biomass did not vary significantly between them.

In summer, heterotrophic bacterioplankton are generally more abundant in the CDW of the

ECS than in other regions (Chen et al., 2006; Chen et al., 2009). Chen et al. (2006) suggested that
the growth of bacteria along the coast might be stimulated by the substantial amount of organic
matter derived from both autochthonous marine production and fluvial runoff. This spatial
distribution pattern was also observed in 2009 and 2010. In the 2009 non-flooding period, the
mean values of the bacterial biomass in the surface water of the CDW and all other areas were
77.5 ($\pm$ 55.7) and 31.0 ($\pm$ 18.6) mg C m$^{-3}$, respectively. Their mean values in the 2010 flood were
24.4 ($\pm$ 18.6) and 15.0 ($\pm$ 11.5) mg C m$^{-3}$ in the CDW and other regions, respectively. Further
analyses revealed that the bacterial biomass in the surface water was positively and linearly
associated with Chl *a* concentrations in both 2009 ($p < 0.01$) and 2010 ($p < 0.05$). This finding
applies to the values averaged over $Z_E$ in both periods (both $p < 0.01$). These results suggest that
in both study periods, bacterial growth might have been associated with the organic carbon



derived from phytoplankton. However, the mean values of Chl *a* concentrations in the surface
water were slightly higher in 2010 than in 2009 (Table 1).

In general, an increased amount of organic matter is delivered through fluvial discharge into

the ECS during periods of high riverine flow (e.g., Wang et al., 2012). Although these results
suggest that the bacterial biomass might be higher in the flooding period than in the non-flooding
period, this difference was not verified when using averaged bacterial biomass values in this
study. The bacterial biomass in the surface water was significantly higher in the 2009 non-
flooding period than during the 2010 flood, with mean values of 39.8 ($\pm$ 33.7) and 20.4 ($\pm$ 16.5)
mg C m$^{-3}$, respectively (Table 1). The average bacterial value over $Z_E$ was even more pronounced
in 2009 than in 2010 (data not shown). However, the total bacterial biomass in the CDW zone
was two times higher in 2010 than in 2009, with values of 47.7 and 21.0 x 10$^6$ kg C, respectively
(Table 2).

In addition, the major taxa of bacterioplankton varied between both periods (C.C. Chung,

unpublished data). During the non-flooding period, cyanobacteria were predominant (70% of the
bacterioplankton community) at the selected sampling stations located either in the CDW zone or
in other regions of the ECS. During the 2010 flood, such a high percentage of cyanobacteria was
observed only at stations located in regions other than the CDW. At this survey time, in addition
to cyanobacteria, the dominant taxa of bacterioplankton in the CDW zone also included



favobacteria, gammabacteria, alphabacteria, and actinobacteria. A potential cause of the low
average bacterial biomass observed during the 2010 flood might be protozoan grazing. Protozoa
have been recognized as important microbial grazers in the ECS and in many coastal ecosystems
(e.g., Chen et al., 2009; Chen et al., 2003; Sherr and Sherr, 1984). Although protozoan abundance
was not measured in this study, a high production rate of nanoflagellates was observed in the
southern ECS, with mean values of 0.46 µg C $l^{-1}$ $h^{-1}$ during periods of high riverine flow (Tsai et
al., 2005).

Zooplankton are amongst the most important contributors to plankton CR (Calbet and

Landry, 2004; Hernández-León and Ikeda, 2005; Hopkinson Jr. et al., 1989). In this study,
zooplankton were only sampled across the whole water column. However, the average biomass
of zooplankton over $Z_E$ can be still estimated. The zooplankton biomass over $Z_E$ was significantly
higher in the 2010 flood than in the 2009 non-flooding period, with mean values of 105.7 (±
144.4) and 22.6 (± 25.7) mg C $m^{-3}$, respectively ($p < 0.01$). The average zooplankton biomass
over $Z_E$ for the CDW zone was 90-fold higher in 2010 than in 2009 (Table 2), suggesting that the
flood may have had a significant effect on zooplankton biomass. The high zooplankton biomass
observed in 2010 also implies that plankton communities might be in the late phase of succession
during this flood event.

**Effects of the Changjiang River flooding on plankton community respiration**



Plankton CR has been assumed to be an integrated rate of organic carbon consumption by
plankton communities (e.g.., Hopkinson Jr. et al., 1989; Rowe et al., 1986). In summer, the mean
CR rate in the surface water of the ECS ranges from 52.2 to 128.4 mg C m$^{-3}$ d$^{-1}$ (Chen et al.,
2006; Chen et al., 2009). The CR rate has been significantly correlated with fluvial discharge
from the Changjiang River (Chen et al., 2009). In this study, the CR in the surface water ranged
from 2.7 to 311.9 mg C m$^{-3}$ d$^{-1}$, with a mean value of 73.2 ($\pm$ 76.9) mg C m$^{-3}$ d$^{-1}$ in the 2009 non-
flooding period (Table 1). During the 2010 flood, this rate in the surface water was significantly
higher than in 2009 ($p < 0.01$; Table 1). The value of CR in the surface water was in the range of
10.9-325.3 mg C m$^{-3}$ d$^{-1}$, with a mean value of 105.6 ($\pm$ 66.7) mg C m$^{-3}$ d$^{-1}$ (Table 1). The CR rate
averaged over the $Z_E$ was also higher in 2010 than in 2009, with mean values of 76.8 ($\pm$53.0) and
66.8 ($\pm$68.4) mg C m$^{-3}$ d$^{-1}$, respectively. However, the difference was not statistically significant
($p = 0.08$). In terms of spatial distribution, higher CR rates were mostly observed in the CDW
region in both sampling periods, especially along the coast (Fig. 2). Nevertheless, it should be
noted that the CDW widely expanded in 2010 compared to during 2009. These results also reveal
that the CR in the summer of the 2010 flood was at the high end of the values typically observed
in the ECS (Chen et al., 2006; Chen et al., 2009), suggesting that the CR might have been
enhanced by the Changjiang River flooding in 2010.
To assess the biotic controls on CR, the rates were regressed against biomass of





phytoplankton, heterotrophic bacteria, and zooplankton, as well as primary production (when
applicable). An analysis of the pooled data in each period show that CR was significantly
correlated with Chl $a$ concentrations and bacterial biomass (both $p < 0.001$; Fig. 3). In both
periods, the linear relationship was also statistically significant between CR and Chl $a$
concentration and between CR and bacterial biomass, both for the surface water and when
averaged over $Z_E$ (all $p < 0.01$). In addition, in the 2010 flood, CR was significantly correlated
with PP in the pooled data, surface water, or when averaged over $Z_E$ (all $p \leqq 0.01$). Compared
with a previous study in the ECS (Chen et al., 2009), CR (g $O_2$ m$^{-3}$ d$^{-1}$) was also scaled as a
power function of PP (g $O_2$ m$^{-3}$ d$^{-1}$) in the 2010 flood, where CR $= 5.78$ PP$^{1.24}$ ($p = 0.001$; Fig. 4).
Note that the exponent (1.24) represents the slope of the log-log transformation. This value is
larger than previous values reported for the ECS (CR $= 0.58$ PP$^{0.46}$) and other coastal ecosystems
in the world (CR $= 1.1$ PP$^{0.72}$) (Chen et al., 2009; Duarte and Agustí, 1998). To support higher
CR, Chen et al. (2009) suggests that external substrates transported from the river into the ECS
might be an important source during the high-flow summer. This finding however suggests that
during the 2010 flood, in addition to allochthonous organic matter, the higher CR rate might have
also been fueled by *in situ* organic carbon production, such as PP. The important contribution of
phytoplankton and/or bacterioplankton to CR has been identified in the ECS, even though its
relative contribution might vary spatially or temporally (Chen et al., 2006; Chen et al., 2009;



Chen et al., 2003). These results suggest that the CR rate might be dominated by phytoplankton
and bacterioplankton in the 2009 non-flooding period. During the 2010 flood, the higher CR
could be attributed to vigorous plankton metabolic activities, especially phytoplankton.

Surprisingly, the mean Chl $a$ value was slightly higher in 2010 than in 2009. In contrast, the

bacterial biomass was significantly lower in 2010 than in 2009 (Table 1). However, the CR rate
was still higher in 2010 than in 2009. To gain greater insight, the differences (i.e., 2010 minus
2009) in the average CR, Chl $a$ concentration, and bacterial biomass over $Z_E$ at the same station
between two periods were compared. The extent of such differences in CR was significantly
related to differences in Chl $a$ concentration ($p < 0.001$) and bacterial biomass ($p < 0.01$; Fig. 5).
The linear relationships were also statistically significant if the values of the differences in the
surface water were applied (all $p < 0.01$; data not shown). Among the positive CR difference
values (i.e., 20 of 33), 15 stations were also characterized by positive differences in Chl $a$
concentrations, but only two stations had positive differences in bacterial biomass. Interestingly,
the stations with positive Chl $a$ concentration difference values were mostly located within the
CDW region in 2010, with the exception of the CDW in 2009. These results suggest that the
higher CR in the 2010 flood might be attributed to phytoplankton. The mean Chl $a$ concentration
was only slightly higher in 2010 than in 2009. However, the phytoplankton assemblage varied
between periods. Therefore, it is reasonable to speculate that the differences in CR rate in both





periods might have been partially caused by variation in the composition of the phytoplankton
communities. Although the CR attributed to different components of the phytoplankton
community was not measured in this study, it was been documented elsewhere (e.g., Lopez-
Sandoval et al., 2014).
In addition, zooplankton might also be amongst the potential contributors to the higher CR
rate observed in 2010 than in 2009. As stated above, the biomass of zooplankton was
significantly higher in 2010 than in 2009. However, the linear relationships between CR and
zooplankton biomass over $Z_E$ were not statistically significant in 2009 or 2010. To further
explore how plankton communities contributed to CR, the CR rate was regressed against total
plankton biomass (i.e., summed biomass of phytoplankton, bacterioplankton, and zooplankton)
for both periods. The linear relationships were significant between CR and total plankton biomass
(mg C m$^{-3}$) over $Z_E$ both in 2009 ($p < 0.001$) and 2010 ($p < 0.01$; Fig. 6). Similarly significant
relationships between CR and total planktonic biomass have also been observed in the summer in
the ECS, and phytoplankton and bacterioplankton might be the most important components
contributing to CR at such times (Chen et al., 2006). In this study, autotrophic plankton biomass
(i.e., phytoplankton) accounted for 41.3% and 45.6% of total planktonic biomass in 2009 and
2010, respectively. As for heterotrophic plankton biomass, bacterioplankton attributed to 38.7%
and 11.3% and zooplankton contributed for 20.0% and 43.1% of total plankton biomass in 2009



and 2010, respectively. This suggests that phytoplankton and bacterioplankton might be the most
important components attributing to CR in the 2009 non-flooding period. In contrast, during the
2010 flood, the CR rate might have been mostly driven by phytoplankton and zooplankton
metabolic activity. Even though, this conclusion was derived from stocks, and the biomass might
not be directly related to the concurrent CR rate. By using physiological and allometric
relationships of variant plankton communities, the individual CR rate of plankton could be
estimated from its stock and significant regression has been found between measured and
estimated rates (Chen et al., 2009). Furthermore, it also should be noted that microzooplankton
might be another important contributor to CR. Unfortunately, it was not measured and could not
evaluate its contribution to CR in this study.

**Implications of community metabolism in the coastal ecosystem**

To evaluate the metabolic balance of the plankton community, the P/R ratio of PP to CR can

be used as an index (e.g., Duarte and Agustí, 1998; Kemp et al., 1997). It also should be noted
that in this study, the P/R ratio might be under-estimated because the values of P (i.e., PP) and R
(i.e., CR) were integrated over $Z_E$ instead of over the entire water column. In the 2010 flood, the
P/R ratio was in the range of 0.11 to 1.33, but it could not be calculated in 2009 since PP was not
measured in this period. Surprisingly, the mean P/R ratio was similar to that in the summer in the
ECS, with a mean value of $0.42 \pm 0.33$ (Chen et al., 2009). This value, however, was much lower



than the P/R ratio reported in other coastal ecosystems (Duarte and Agustí, 1998). This result
implies that a large amount of organic carbon was respired by the plankton community into the
water column during the flooding period. In addition to phytoplankton, zooplankton may have
also contributed significantly to the high CR in 2010. This assumption is supported by the fact
that high zooplankton biomass (mean value = 105.7 mg C m$^{-3}$) was documented during this
period, and zooplankton also accounted for 43.1% of the total plankton biomass. This result also
suggests that in the 2010 flood, the ECS shelf ecosystems were net heterotrophic. A heterotrophic
ecosystem was documented in the ECS in summer and in other seasons (Chen et al., 2006; Chen
et al., 2013; Chen et al., 2003). A low P/R ratio (i.e., < 1) has been widely observed in coastal
ecosystems worldwide (e.g., del Giorgio et al., 1997; Duarte and Agustí, 1998).
A further comparative analysis was conducted to determine whether the CR rate affected the
fugacity of $CO_2$ ($fCO_2$) in the seawater. In 2009, the $fCO_2$ in the surface water was in the range of
118.7-599.8 µatm, with mean values of 362.9 ± 101.2 µatm (Table 1). This mean value is close to
the mean value (369.6 µatm) observed in the ECS in August in prior years (Chen et al., 2006). In
the 2010 flood, the mean value (297.6 µatm) of $fCO_2$ in the surface water was significantly lower
than in 2009, and ranged from 178.7 to 454.2 µatm (Table 1). It is well known that $fCO_2$ is
temperature dependent, and it increases as the temperature increases (e.g., Goyet et al., 1993).
The effect of temperature on the large variation in $fCO_2$ observed between the 2009 non-flooding



period and the 2010 flood might be trivial, though, because the SST was similar in both periods
(Table 1).

The effect of freshwater on $f$CO$_2$ in the surface water in the ECS has also been suggested to

be relatively minor compared to the inter-annual variation of $f$CO$_2$ (Chen et al., 2013). To
evaluate this, conservative mixing was applied by using TA and DIC data between freshwater
and seawater end-members. The TA and DIC data reported by Zhai et al. (2007) for the
Changjiang River in summer were used as freshwater data (both TA and DIC = 1743 μmol kg$^{-1}$).
The surface data at Station K, shown at the bottom right-hand side of Fig. 1, were selected to
represent the seawater data (SSS = 33.96, TA = 2241 μmol kg$^{-1}$, and DIC = 1909 μmol kg$^{-1}$ in
2009; SSS = 33.96, TA = 2240 μmol kg$^{-1}$, and DIC = 1904 μmol kg$^{-1}$ in 2010). The simulated
results show that the effect of mixing freshwater and seawater on $f$CO$_2$ was nearly the same in
both periods. However, a large variation in $f$CO$_2$ in the surface water was estimated; it varied
from 439.8 to 375.4 μatm within a salinity range of 20.38 to 33.96. This finding implies that
surface water $f$CO$_2$ in the ECS might increase dramatically, especially during the devastating
flood of 2010 where low SSS (≤ 31 psu) characterized almost 70% of the ECS shelf (Fig. 1b).

However, in the 2010 flood, surface water with low $f$CO$_2$ was observed in the ECS.

Therefore, vigorous photosynthetic processes might be a potential cause for the reduction of $f$CO$_2$
in the surface water during periods of flooding. Compared to PP values observed in summer in



the ECS in previous years (Chen et al., 2009), primary production was indeed high during the
2010 flood (Table 1; Chen et al., 2009). Gong et al. (2011) also estimated that over the past
decade, the carbon fixation rate during flooding was about three times higher than during non-
flooding periods. However, no significant relationship was found between $f\mathrm{CO_2}$ and PP in the
2010 flood, though this may simply be due to having a small sample size for PP. Nevertheless,
$f\mathrm{CO_2}$ was significantly related to Chl $a$ concentration in the pooled data of the 2010 flood ($p <$
0.001). This significant relationship indirectly supports that the reduction in $f\mathrm{CO_2}$ in the 2010
flood might be associated with vigorous phytoplankton metabolic activity. Furthermore, negative
linear relationships were observed between $f\mathrm{CO_2}$ and CR in the surface water during both the
2009 non-flooding period ($p < 0.01$) and in the 2010 flood ($p < 0.001$; Fig. 7). Significant linear
relationships were also found using pooled data from each period (all $p < 0.001$). CR has been
assumed to be an integrated response of overall plankton activity. These results imply that $f\mathrm{CO_2}$
in the surface water (or water column) is related to plankton activities. To explore the variations
in $f\mathrm{CO_2}$ between the non-flooding and flooding periods, the difference in $f\mathrm{CO_2}$ and CR at the
same station was estimated. Surprisingly, a negative linear relationship was found between the
difference in $f\mathrm{CO_2}$ and CR of the flooding and non-flooding periods ($p = 0.001$; Fig. 8). As
previously stated, compared to the 2009 non-flooding period, the increase in CR rate in the 2010
flood might be associated with the increase in phytoplankton biomass (Fig. 5a). These results



indicate that the significant amount of $f$CO$_2$ absorption in the 2010 flood was related to the
strength of plankton activity, particularly phytoplankton at stations that were not characterized by
low SSS in the 2009 non-flooding period.

**CONCLUSION**

Riverine run-off has a profound effect on organic carbon production and consumption in

coastal ecosystems globally. It has become even more pronounced with the dramatic increase in
extreme rainfall events and flood magnitude in the Changjiang River and around the world. In
July 2010, a devastating flood occurred in the Changjiang River, and this flood started in late
May or early June. This event provided an opportunity to investigate the effects of flooding on
pelagic ecosystem and plankton community respiration (CR) in the ECS shelf. A comparative
analysis was conducted between the data obtained during this flood versus those gathered in July
2009, when the riverine flow was relatively low. During the flood, a large amount of freshwater
was discharged into the ECS. The CDW zone, where SSS ≤ 31 psu, covered almost two thirds of
the continental shelf. In the 2010 flood, the CDW zone was approximately six times larger than
in the 2009 non-flooding period.

Higher nitrate concentrations, mostly in the fluvial discharge of the Changjiang River, were

also measured in the ECS during the flood. The comparison of both periods showed that the
nitrate concentration in the surface water of the ECS was significantly higher in the 2010 flood



than in the 2009 non-flooding period. Nevertheless, the phytoplankton biomass in the surface
water or averaged over the $Z_E$ showed no significant difference between 2009 and 2010. An
effect of flooding on phytoplankton biomass was still observed when using total Chl *a* content
over $Z_E$ integrated for the entire ECS or the CDW zone, and the total Chl *a* content in the ECS
was higher in 2010 than in 2009. In addition, in the 2010 flood, PP in the surface water was
relatively high when compared to PP observed in prior works. Gong et al. (2011) estimated that
the average rate of carbon fixation during the flood was $176.0 \times 10^3$ tons C $d^{-1}$, which was about
three times higher than during non-flooding periods over the past decade.

Phytoplankton abundance was twice as high in the CDW than in other regions. In the 2010

flood, the phytoplankton communities predominantly consisted of diatoms; this is in contrast to
assemblages documented during non-flooding periods, in which dinoflagellates,
coccolithophorids, and green algae were also common (Chien, 2009). Surprisingly, the bacterial
biomass in the surface water was significantly higher in the 2009 non-flooding period than in the
2010 flood. Despite this, CR was still higher during the 2010 flood than in the 2009 non-flooding
period, with mean $\pm$ SD values of $105.6 \pm 66.7$ and $73.2 \pm 76.9$ mg C $m^{-3}$ $d^{-1}$ in the surface water,
respectively. The 2010 flood minus the 2009 non-flooding period difference in CR was
significantly related to the respective differences in Chl *a* concentration, suggesting that higher
CR in the 2010 flood might have been attributed to a higher biomass of phytoplankton, especially



in stations located within the CDW region (most of which were not characterized by low SSS in
the 2009 non-flooding period). In addition to phytoplankton, zooplankton might be another
important component contributing to the high CR rate observed in the 2010 flood. This could be
evidenced from the fact that zooplankton biomass in 2010 accounted for 43.1% of the total
plankton biomass (i.e., summed biomass of phytoplankton, bacterioplankton, and zooplankton).

Finally, a negative linear relationship was found between the differences (i.e., 2010 minus

2009) in CR vs. $f$CO$_2$. This finding implies that a tremendous amount of $f$CO$_2$ was absorbed by
vigorous photosynthetic activity during the flood period. Overall, these results suggest that
plankton activity flourished in the substantial amount of dissolved inorganic nutrients discharged
by the river during the flood. This effect was especially pronounced at stations not previously
characterized by low SSS, indicating that the effects of flooding on the ECS shelf ecosystem
might be scaled to the magnitude of the flood.



ACKNOWLEDGEMENTS

This study is part of the multidisciplinary "Long-term Observation and Research of the East

China Sea" (LORECS) and "Effects of Global Chang on Ocean Biogeochemistry and Ecosystem
in the Sea surrounding Taiwan in the Northwest Pacific" (ECOBEST) programs, which are
supported by Taiwan's Ministry of Science and Technology (MOST), ROC under grants NSC-
98-2611-M-003-001-MY3 and MOST 104-2611-M-003-001 to C.-C. Chen. The work of G.-C.
Gong was partly supported by the Center of Marine Bioscience and Biotechnology, National
Taiwan Ocean University. We are furthermore indebted to the officers and crew of the *Ocean
Researcher I* for their assistance during the research cruise. The authors are also grateful to Prof.
T. C. Malone at Horn Point Laboratory at UMCES for providing valuable and constructive
comments to improve the manuscript. Finally, we would like to thank Dr. Anderson Mayfield for
his assistance in English proofing of the final version of the manuscript. This article was
subsidized by National Taiwan Normal University (NTNU), Taiwan, ROC.




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



Table 1. Range of values for different variables in the surface water of the ECS during non-
flooding (2009) and flood (2010) periods, with mean ± SD in parentheses and in brackets
for all sampling stations and for stations in the area of the Changjiang Diluted Water
(CDW) region, respectively. Variables include transparency ($CTD_{TM}$; %), salinity (SSS;
psu), temperature (SST; ℃), fugacity of $CO_2$ ($fCO_2$; µatm), nitrate ($NO_3^-$; µM),
phosphate ($PO_4^{3-}$; µM), silicate ($SiO_4^-$; µM), chlorophyll $a$ (Chl $a$; mg Chl m$^{-3}$), bacterial
biomass (BB; mg C m$^{-3}$), primary production (PP; mg C m$^{-3}$ d$^{-1}$), and plankton
community respiration (CR; mg C m$^{-3}$ d$^{-1}$). For reference, values of euphotic depth ($Z_E$;
m) and mixed layer depth ($M_D$; m) are also shown. The Mann-Whitney Rank Sum test
was applied for variable comparisons between 2009 and 2010, and the results are
indicated as described in the table footnotes.

| Variable | 2009 | 2010 |
|---|---|---|
| $Z_E$ | 1.3–190.6 (38.9±36.4) [16.8±7.4] | 10.1–82.2 (33.4±17.3) [24.8±10.7] |
| $M_D$ | 5–37 (13.7±7.3) [7.3±3.6] | 4–35 (11.3±6.6) [7.9±2.6] |
| $CTD_{TM}$ | 37.2–86.3 (76.7±12.2) [70.0±4.9] | 67.7–88.5 (80.5±5.4) [78.4±4.3]** |
| SSS | 23.80–34.11 (32.62±2.07) [29.24±2.52] | 19.33–34.27 (30.32±3.60)* [27.95±3.03] |
| SST | 23.3–29.6 (26.8±1.7) [25.0±0.9] | 21.0–30.0 (26.1±2.2) [25.1±1.7] |
| $fCO_2$ | 118.7–599.8 (362.9±101.2) [230.4±105.3] | 178.7–454.2 (297.6±79.0)* [248.6±54.5] |
| $NO_3^-$ | 0.0–24.3 (2.0±5.3) [4.0±9.1] | 0.0–37.6 (6.2±9.8)* [10.3±11.3]* |
| $PO_4^{3-}$ | 0.00–0.83 (0.13±0.17) [0.13±0.07] | 0.00–1.71 (0.17±0.30) [0.23±0.37] |





| | | |
|---|---|---|
| SiO$_4^-$ | 1.5–24.5 (5.8±5.9) [9.8±7.2] | 0.6–36.4 (6.4±7.8) [9.1±9.2] |
| Chl $a$ | 0.12–4.41 (0.98±1.52) [2.23±1.46] | 0.03–5.32 (1.26±1.27) [1.83±1.35] |
| BB | 10.6–184.8 (39.8±33.7) [54.9±39.6] | 3.6–90.2 (20.4±16.5)** [24.4±18.6]** |
| PP | – – | 10.0–111.3 (62.1±33.8) [71.0±29.1] |
| CR | 2.7–311.9 (73.2±76.9) [172.0±109.2] | 10.9–325.3 (105.6±66.7)* [142.0±61.2] |

–: no data; *: $p < 0.01$; **: $p < 0.001$




Table 2. Total biomass of biological variables over the euphotic depth integrated for the whole
ECS and the Changjiang Diluted Water region (in parentheses) during non-flooding
(2009) and flooding (2010) periods. Variables include chlorophyll $a$ (Chl $a$; x $10^6$ kg
Chl), bacterial biomass (BB; x $10^6$ kg C), and zooplankton (Zoo; x $10^6$ kg C). For
reference, the areas (x $10^3$ km$^2$) of the entire ECS and CDW regions are also shown.

| Variables | 2009 (non-flooding period) | 2010 (flood) |
| --- | --- | --- |
| Area | 186.0 (19.0) | 182.7 (111.7) |
| Chl $a$ | 4.4 (1.2) | 5.5 (3.9) |
| BB | 222.5 (21.0) | 87.3 (47.7) |
| Zoo | 410.3 (6.2) | 920.6 (560.8) |






FIGURE LEGENDS

Fig. 1. Contour plots of salinity (SSS), nitrate ($NO_3^-$), phosphate ($PO_4^{3-}$), and chlorophyll $a$ (Chl

$a$) in the surface water (2-3 m) in the ECS during non-flooding (2009; left most panels)

and flooding (2010; right-most panels) periods. Bottom depth contours are shown as

dashed lines, both here and in Fig. 2. The sampling stations in both periods are marked by

an ex (x), both here and in Fig. 2. The contour intervals of SSS, nitrate, phosphate, and

Chl $a$ are 0.5 psu, 1.0 µM, 0.1 µM, and 0.5 mg Chl m$^{-3}$, respectively. For reference, the

contour lines (bold) of SSS = 31 psu, $NO_3^-$ = 3.0 µM, $PO_4^{3-}$ = 1.0 µM, and Chl $a$ = 1.0 mg

Chl m$^{-3}$. The range for each parameter is shown at the top of each panel.

Fig. 2. Contour plots of plankton community respiration (CR; mg C m$^{-3}$ d$^{-1}$) over the euphotic

zone of the ECS during a) non-flooding (2009) and b) flooding (2010) periods. The

contour interval is 10 mg C m$^{-3}$ d$^{-1}$. The CR range is shown at the top of each panel.

Fig. 3. Relationships between plankton community respiration (CR; mg C m$^{-3}$ d$^{-1}$) and a)

chlorophyll $a$ concentration (Chl $a$; mg Chl m$^{-3}$) and b) bacterial biomass (mg C m$^{-3}$) for

all data from non-flooding (2009; ●) and flooding (2010; ○) periods. Linear regressions of

data from 2009 (solid lines) and 2010 (dashed lines), as well as the respective $r^2$ and $p$

values, have also been included.

Fig. 4. Log-log relationships between averaged volumetric rates of primary production (PP,

converted to $O_2$ units) and volumetric rates of CR in 2010 (●). Please note the log scale of

both axes. The solid line shows the relationship as a power function of PP. For

comparison, the estimated power function of CR versus PP (CR = 0.58 PP$^{0.46}$) in summer

(○) in the ECS is shown as a dashed gray line (Chen et al., 2009).

Fig. 5. Differences (Δ) between 2010 and 2009 in plankton community respiration (CR; mg C m$^{-3}$

d$^{-1}$) versus a) chlorophyll $a$ (Chl $a$; mg Chl m$^{-3}$) and b) bacterial biomass (mg C m$^{-3}$) over





the euphotic zone at the same station. The $r^2$ and $p$ values have been shown for the best-fit
linear regression line (solid line). For reference, the vertical and horizontal dashed lines
represent inter-year differences of zero (i.e., $\Delta = 0$).
Fig. 6. Relationship between plankton community respiration (CR) and total plankton biomass
(expressed per carbon unit) over $Z_E$ in 2009 (●; solid line) and 2010 (○; dashed line). The
respective $p$ and $r^2$ values are shown for each linear regression line. Total plankton
biomass was the summed biomass of phytoplankton, bacterioplankton, and zooplankton.
Please refer to the "Materials and Methods" for details of the carbon conversion for
plankton communities.
Fig. 7. Relationships between the fugacity of $CO_2$ ($fCO_2$) and plankton community respiration
(CR) in the surface water in 2009 (●; solid line) and 2010 (○; dashed line). The respective
$p$ and $r^2$ values are shown for each linear regression line.
Fig. 8. Differences ($\Delta$) between 2010 and 2009 in $fCO_2$ (µatm) and plankton community
respiration (CR; mg C m$^{-3}$ d$^{-1}$) in the surface water at the same station. For reference, the
vertical and horizontal dashed lines represent the inter-annual differences of zero (i.e., $\Delta =$

0).





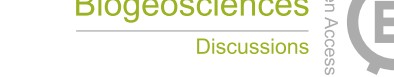


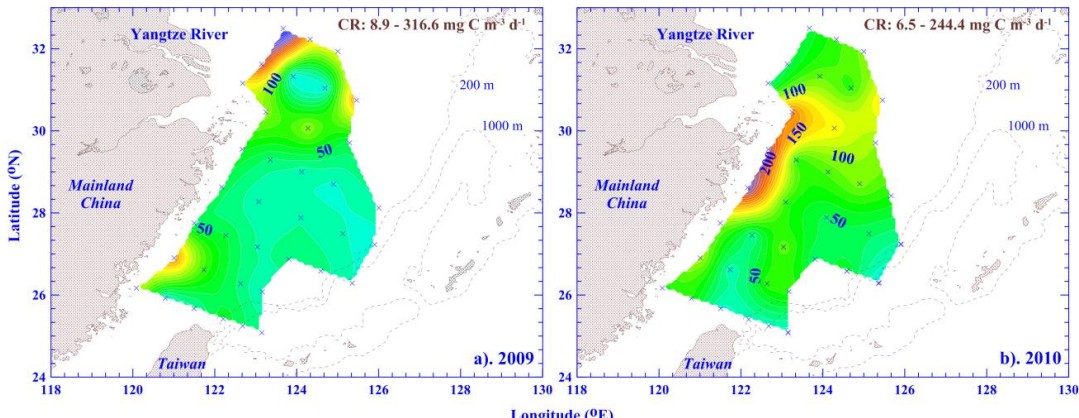


Fig. 2. Contour plots of plankton community respiration (CR; mg C m$^{-3}$ d$^{-1}$) over the euphotic

zone of the ECS during a) non-flooding (2009) and b) flooding (2010) periods. The

contour interval is 10 mg C m$^{-3}$ d$^{-1}$. The CR range is shown at the top of each panel.




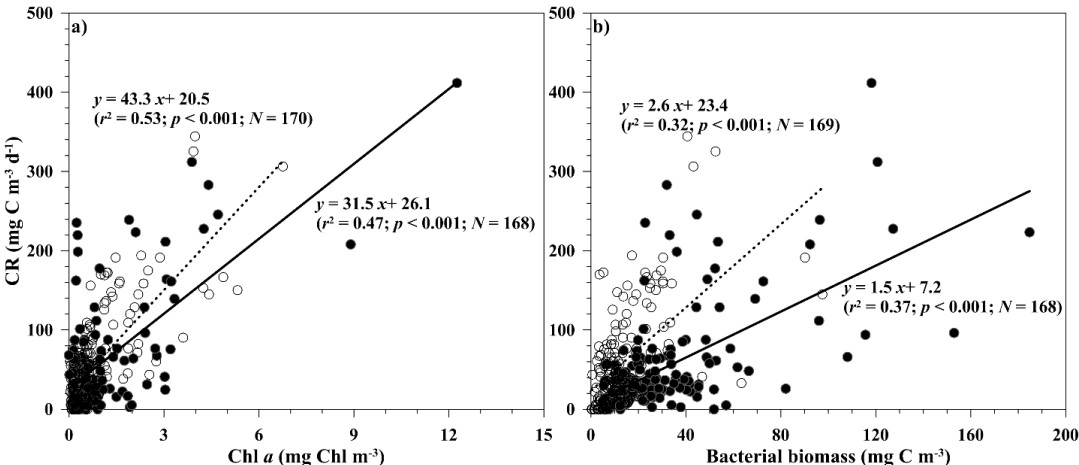


Fig. 3. Relationships between plankton community respiration (CR; mg C m$^{-3}$ d$^{-1}$) and a)

chlorophyll *a* concentration (Chl *a*; mg Chl m$^{-3}$) and b) bacterial biomass (mg C m$^{-3}$) for

all data from non-flooding (2009; ●) and flooding (2010; ○) periods. Linear regressions of

data from 2009 (solid lines) and 2010 (dashed lines), as well as the respective $r^2$ and $p$

values, have also been included.



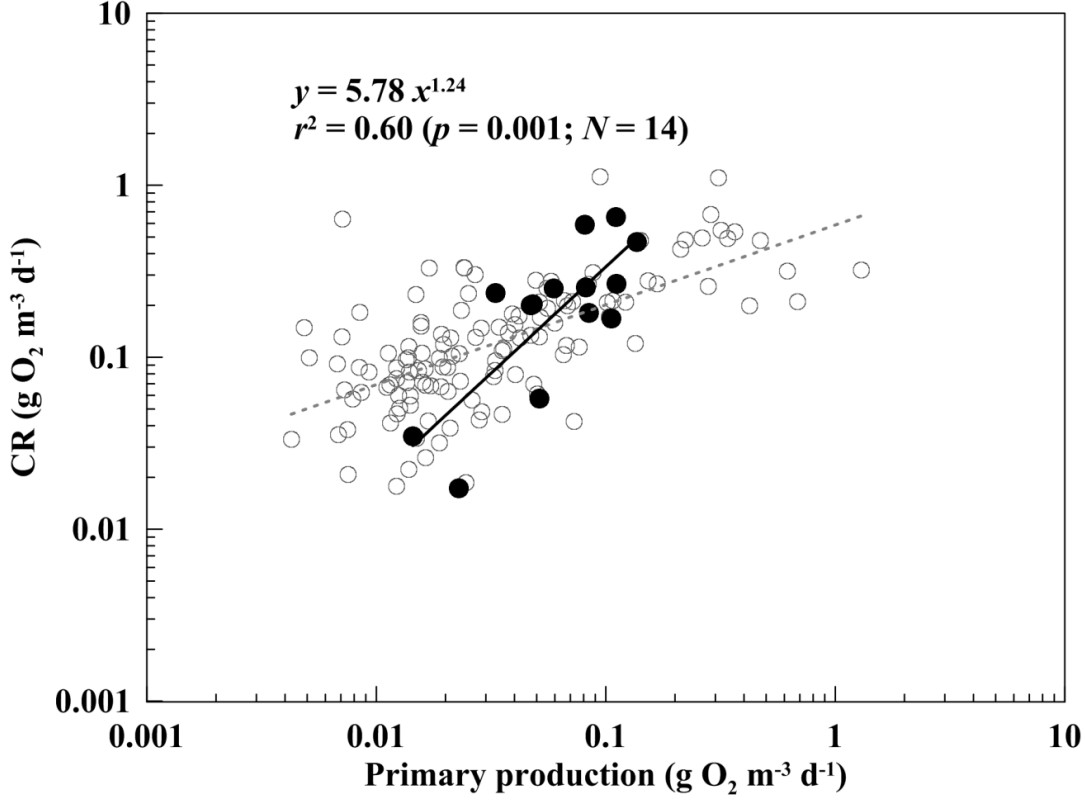

Fig. 4. Log-log relationships between averaged volumetric rates of primary production (PP,

converted to $O_2$ units) and volumetric rates of CR in 2010 (●). Please note the log scale of

both axes. The solid line shows the relationship as a power function of PP. For

comparison, the estimated power function of CR versus PP (CR = 0.58 $PP^{0.46}$) in summer

(○) in the ECS is shown as a dashed gray line (Chen et al., 2009).



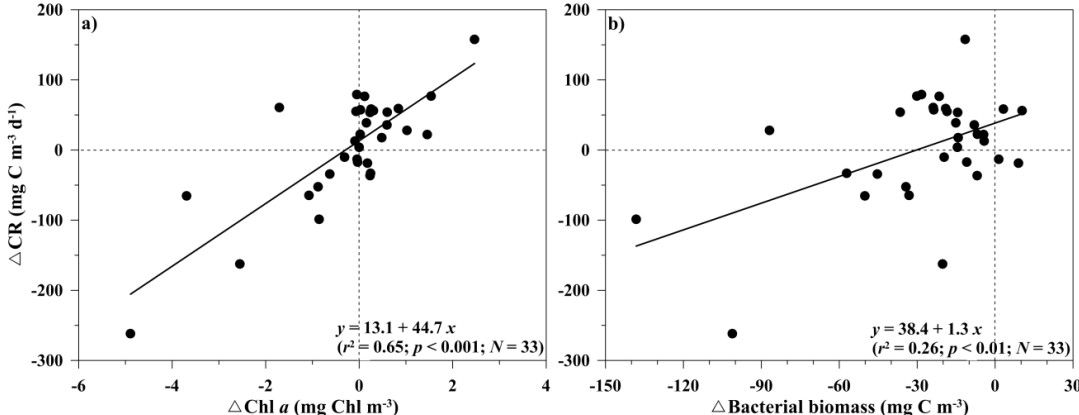

Fig. 5. Differences (Δ) between 2010 and 2009 in plankton community respiration (CR; mg C m$^{-3}$

d$^{-1}$) versus a) chlorophyll *a* (Chl *a*; mg Chl m$^{-3}$) and b) bacterial biomass (mg C m$^{-3}$) over

the euphotic zone at the same station. The $r^2$ and $p$ values have been shown for the best-fit

linear regression line (solid line). For reference, the vertical and horizontal dashed lines

represent inter-year differences of zero (i.e., Δ = 0).




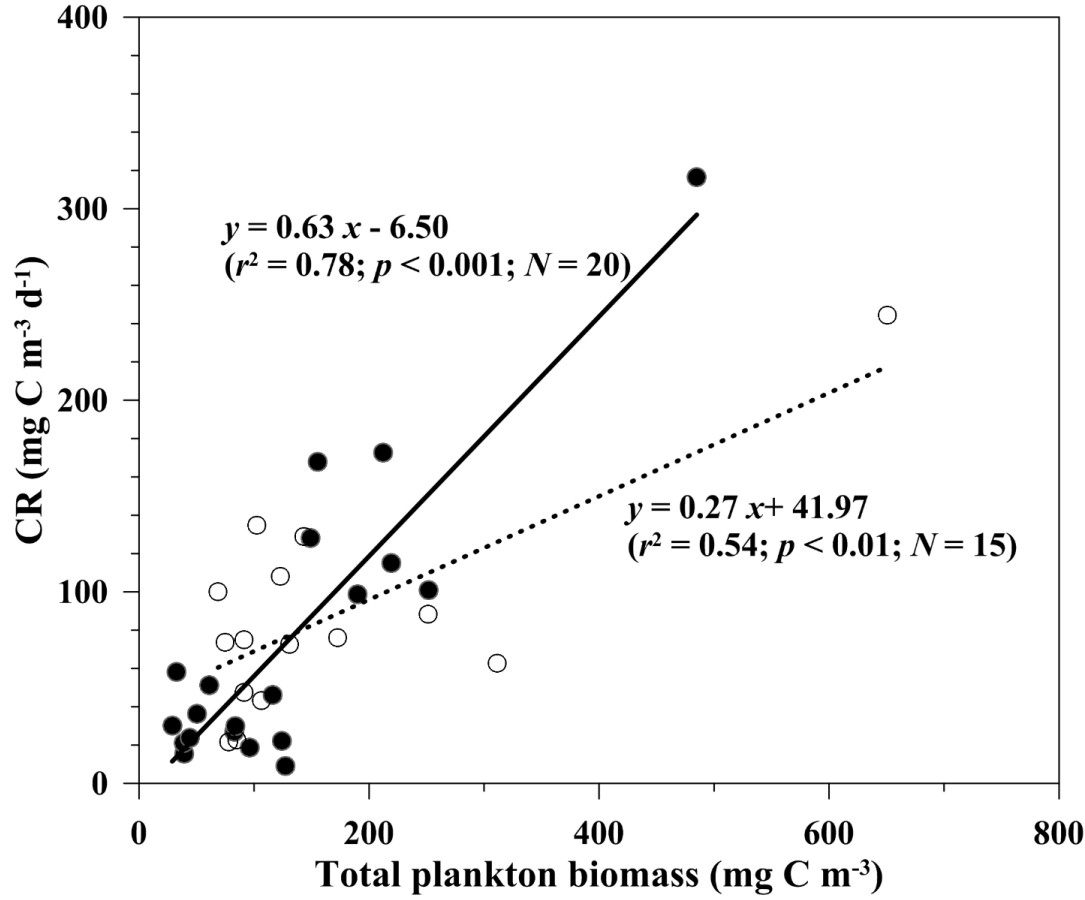


Fig. 6. Relationship between plankton community respiration (CR) and total plankton biomass
(expressed per carbon unit) over $Z_E$ in 2009 (●; solid line) and 2010 (○; dashed line). The
respective $p$ and $r^2$ values are shown for each linear regression line. Total plankton
biomass was the summed biomass of phytoplankton, bacterioplankton, and zooplankton.
Please refer to the "Materials and Methods" for details of the carbon conversion for
plankton communities.





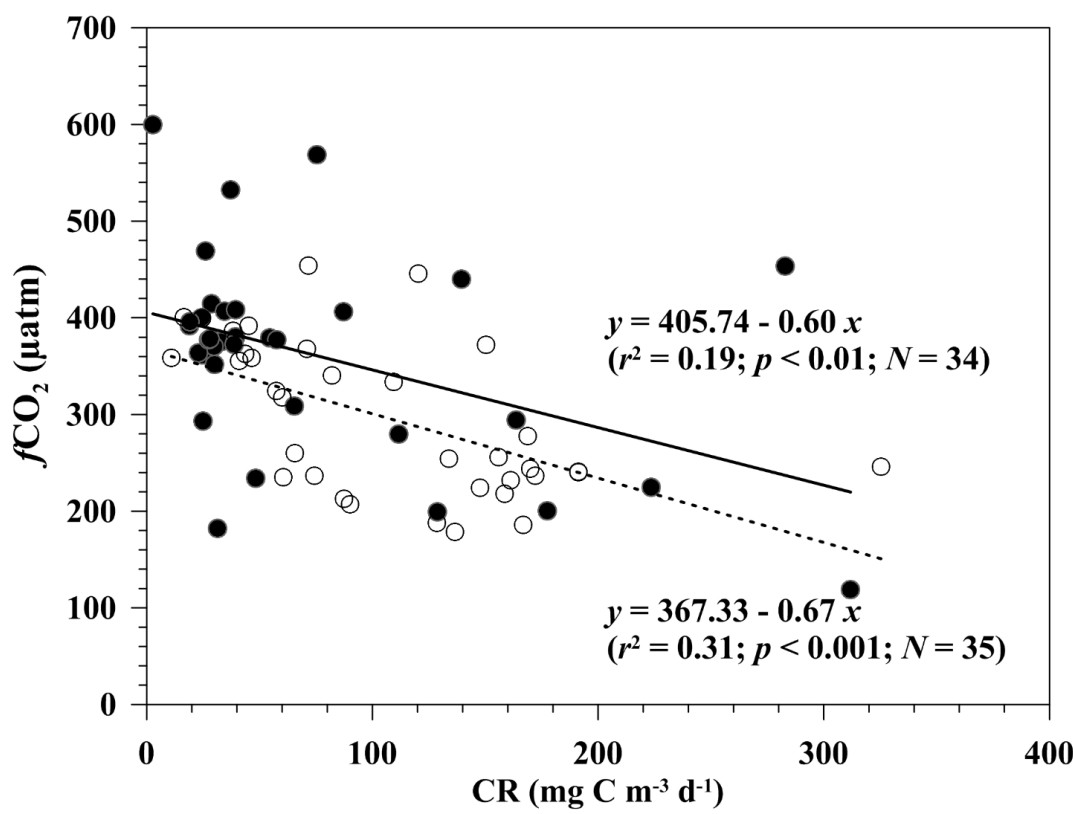


Fig. 7. Relationships between the fugacity of $CO_2$ ($fCO_2$) and plankton community respiration

(CR) in the surface water in 2009 (●; solid line) and 2010 (○; dashed line). The respective

$p$ and $r^2$ values are shown for each linear regression line.



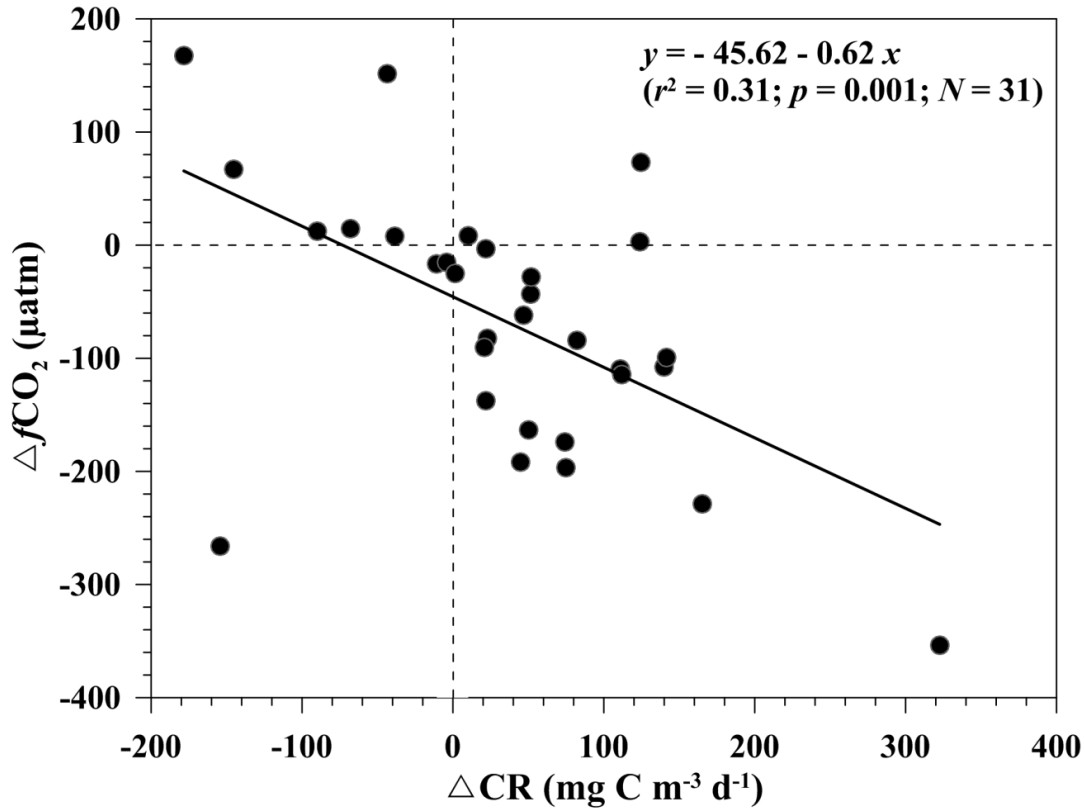


Fig. 8. Differences (Δ) between 2010 and 2009 in $f$CO$_2$ (μatm) and plankton community

respiration (CR; mg C m$^{-3}$ d$^{-1}$) in the surface water at the same station. For reference, the

vertical and horizontal dashed lines represent the inter-annual differences of zero (i.e., Δ =

0).