# Peer review of "The influence of episodic flooding on pelagic ecosystem in the East China Sea"

_Biogeosciences, 2016_

## Referee Comment (RC1) · Anonymous Referee #1 · 6 Aug 2016

General Comments: This study presents an interesting analysis of pelagic ecosystem responses in the East China Sea to the large flood event in the Changjiang River in 2010. This research is fundamentally focused on "an episodic event", so it would not be too surprising if the scope of relevant measurement is limited. Nonetheless, the authors conducted a fairly good job of synthesizing what they have learned based on their current and others' previous observations and various indices and metrics (e.g. volumetric values in surface water, averaged over the depth of euphotic zone, and depth-integrated values for the entire ECS and the Changjiang Diluted Water (CDW) region). Overall, the analysis is thorough, but I still have a few major suggestions and revisions in terms of statistics/statistical interpretation to suggest before I think this study is ready for publication. These include the status of nutrient limitations based on significant regression relationships (below comment #8) and the relative strength

of coupling/control of one variable to another with the regression slope values (below comment #10). I also suggest authors to clarify their calculations on estimating the effect of freshwater discharge on fCO2 using end-member mixing equations. Below are a range of suggestions, questions, and comments that should be addressed in the revision. Overall, I think this study would be of high interest to the readership after substantial revisions.

Specific Comments: 1. Abstract, line 43: The sentence "... which were not characterized by low SSS in 2009" is not correct. Table 1 indicates SSS in CDW zone was also lower than the entire area (the area for all sampling stations) in 2009. 2. Keywords, line 51: It would be better to be more specific. Perhaps you could add "flooding", "CDW", "freshwater discharge" and also "Yangtze River"? 3. Comments on Figure 1: 1) The color of the SSS contour plot is confusing to read. Usually with salinity contours, the bluer the fresher and the redder the saltier, 2) you could add the color bars for Figures 1 and 2 both, and 3) please increase the font size. 4. Comments on Table 1: 1) I think it is critical to compare if there are significant differences between the entire area and CDW zone for all variables in each year as well as overall all 2 years combined, and 2) why zooplankton values are not reported? They are in Table 2, so it should be able to calculate them. 5. Comments on Table 2: 1) To be consistent with Table 1, you could use brackets for values in the CDW zone instead of parentheses. Also, in Table 1 you could report average values for the entire area (what are in parentheses for now) without parentheses and their ranges in parentheses instead (e.g. SSS: 32.62+/-2.07 (23.80-34.11)). 6. Results & discussion, line 226: Please clarify and write more clearly if 2003 was also an anomalous flooding year and how does its area of CDW zone compare with that in 2010. 7. Results & discussion, line 235: Add the sentence what the response time of phytoplankton bloom is to flooding events if reference exists. 8. Results & discussion, line 272: I think you need to be cautious about relying solely on N/P elemental ratio for discussing the status of nutrient limitation. The fact that during the 2010 flood chlorophyll was positively correlated with nitrate and silicate (lines 290) but not to (at alpha = 0.05) phosphate (line 291) suggests limitations of nitrate and silicate predominantly, and to a lesser extent, of phosphate on phytoplankton productivity. This is opposite to your statement that 2010 was more likely affected by phosphate limitation. Similarly, phytoplankton might be limited by all three nutrients in 2009 (lines 294-296). Other suggestion is that it would better to focus only on the area in which nitrate and phosphate values were below detection limits as indicated in Table 1. 9. Results & discussion, line 413-414: I do not follow. Please see below comments on Figures 3-5. 10. Comments on Figures 3, 4, 5: When two variables are regressed onto each other, the slope of regression could be used to assess the strength of coupling between the two or control of one to another – the higher slope, the strong coupling/control, where even a small change in X-values results in a huge response in Y-values. In this regard, for Figure 3 I would consider stronger control of chlorophyll and bacterial biomass on CR in 2010 than in 2009 if you do inter-year comparisons. Figure 4 might also be similarly interpreted (the stronger control of PP on CR in 2010 than in 2009). The same goes for Figure 5. Additionally, if the unit of chlorophyll is converted to that of bacterial biomass, the relative strength of control by each on CR could also be inferred. 11. Results & discussion, line 468: This is only true relative to global average values in coastal oceans. I do not think you could say this unless you compare with P/R value in the non-flooding year, which you do not have the data for. 12. Results & discussion, lines 487-499: I suggest you provide a full summary of values calculated from endmember mixing equations, including equations and how to calculate uncertainties, RMSE, etc. Also, you might want to more emphasize that the calculations presented in this paragraph give you estimates that are expected by purely physical processes, without taking biological effect into account. Deviation from what is estimated based on endmember mixing model is due to biological effect, which you nicely described in the following paragraph.

Technical Corrections: 1. Line 129: Replace "estimated" with "calculated" 2. Line 159: Replace "multiple" with "multiplied" 3. Line 189: Avoid subjective words like "devastating". 4. Line 240: "immense" to "large" 5. Line 264: Replace "16" with canonical Redfield ratio for N:P of 16 or so. 6. Line 418: Awkward phrase "to gain greater insight".

7. Lines 433-444: Please be specific. 8. Line 463: I think it is "overestimated" given the vertical profile of P and R? 9. Line 484-486: Please be specific and give numbers from the calculation.

---

## Referee Comment (RC2) · Anonymous Referee #2 · 4 Sep 2016

Review of Chung-Chi Chen et al submitted to Biogeosciences The aim of this paper is stated to 'reveal the effects of riverine input of dissolved inorganic nutrients on the plankton communities that support heterotrophic processes in the East China Sea shelf ecosystem between periods of non-flooding and flooding.

Generally the topic of the paper is clearly introduced as a comparison of data collected during summer surveys of the ECS in July 2009 and 2010 with 2010 being a year when exceptional river flows from the Changjiang river impacted the coast waters of the ECS.

The methods are reasonably clearly described with references to several previous papers by the research team. However the collection of zooplankton needs more explanation – if they were vertical hauls through the water column give the depth range. Were the zooplankton preserved in formalin prior to counting?....Also it is rather non-

standard to use GF/F filters to collect 14C labelled phytoplankton following incubations. How significant was the loss of small phytoplankton ie <1um on the 14C uptake rates. Also as this 14C data was only collected during the 2010 survey I suggest it could be removed from the paper. Determining oxygen respiration rates from dark incubation of enclosed water samples by difference between initial fixed samples and final incubated bottles using the Winkler method to analyse for dissolved oxygen is a standard approach. However based on only two initial and two final replicates I suggest will yield low precision measurements. It is standard practise to use at least 4 replicates of initial and final bottle measurements. The precision stated is only really the difference divided by the mean of two replicates and I would suggest rather unreliable.

My main problem with this paper however is the section labelled Results and Discussion. This section of the paper is 18 pages long! If the paper is to be resubmitted I strongly recommend that the results and discussion are presented as two different sections and the discussion section greatly shortened. The discussion and interpretation of the data currently included in the paper is at best speculative and in many places vague with the word 'might' used very frequently in numerous sentences. For example Page 19 lines 326-328 Page 19 line 340 Page 21 lines 323-375 Page 22 line 392 plus many more scattered throughout this section.

The conclusion section also needs to be much shorter and report the studies main findings without including too many references to other studies. In summary I strongly recommend this paper only be considered for publication if following resubmission the results and discussion are rewritten as separate sections and the discussion is greatly shortened and written less speculatively. Specific Comments Page 2 line 42; 'vigorous plankton metabolic activities especially phytoplankton ' – rather vague- be more specific eg respiration? Production? Page 2 line 43 define 'SSS' page 2 line 44 '...zooplankton might be ...' far too vague in abstract. Page 5 line 72 line avoid using the word 'tremendous' Page 5 line 78 and elsewhere delete 'psu' salinity has no units now. Page12 line 211 'previously documented values' – be more specific ie when?

Page 13 line 230 change 'trailing' to 'previous' Page 15 line 261 the single high phosphate concentration also evident on figure 1 looks to be an analytical anomaly. Page 17 line 304 and table 2 data. I do not believe it is useful or that accurate to estimate the total chlorophyll a etc in the ECS. I suggest deleting table 2. Page 46 and 47 Figure 1 and 2. The contour plots are not very clear. The sampling locations need to be more clearly indicated by lager clear symbols.

Figure 3 Although the relationships shown apparently are significant- the considerable scatter is not very convincing. If the one high chlorophyll point is removed from figure 3a is the relationship still significant? The relationships might be more usefully illustrated if the data from each year is shown on separate plots ie 2009 in upper figure and 2010 on lower figure with axis ranges the same on both figures.

―――――――――――――――――――――

---

## Author Comment (AC1) · 28 Sep 2016

**Responses to reviewers' comments on ms no: bg-2016-246 "The influence of episodic flooding on pelagic ecosystem in the East China Sea" (Chen, Gong, Chou, Chung, Hsieh, Shiah, and Chiang)**

**Referee #1**

**General comments:**

*This study presents an interesting analysis of pelagic ecosystem responses in the East China Sea to the large flood event in the Changjiang River in 2010. This research is fundamentally focused on "an episodic event", so it would not be too surprising if the scope of relevant measurement is limited. Nonetheless, the authors conducted a fairly good job of synthesizing what they have learned based on their current and others' previous observations and various indices and metrics (e.g. volumetric values in surface water, averaged over the depth of euphotic zone, and depth-integrated values for the entire ECS and the Changjiang Diluted Water (CDW) region). Overall, the analysis is thorough, but I still have a few major suggestions and revisions in terms of statistics/statistical interpretation to suggest before I think this study is ready for publication. These include the status of nutrient limitations based on significant regression relationships (below comment #8) and the relative strength of coupling/control of one variable to another with the regression slope values (below comment #10). I also suggest authors to clarify their calculations on estimating the effect of freshwater discharge on fCO2 using end-member mixing equations. Below are a range of suggestions, questions, and comments that should be addressed in the revision. Overall, I think this study would be of high interest to the readership after substantial revisions.*

> Thank you so much for your positive evaluation on our manuscript. We are pleased to see that you generally appreciated our presentation of the results. Indeed, this manuscript intends to explore how episodic flooding affects pelagic ecosystems in the East China Sea. As you commented, the results should be of interest to many readers since extreme rainfall events are predicted to increase globally due to climate change. We drew upon a comprehensive dataset to make our conclusions. We have substantially revised our manuscript following your recommendations, as well as those of the other reviewers and appreciate the constructive valuable comments we received. Please see our detailed responses to your comments below. We hope that we have sufficiently addressed your concerns, as they have greatly improved the manuscript. We are now confident that this manuscript is suitable for publication in *Biogeosciences*.

**Specific Comments:**

*1. Abstract, line 43: The sentence ": : : which were not characterized by low SSS in 2009" is not correct. Table 1 indicates SSS in CDW zone was also lower than the entire area (the area for all sampling stations) in 2009.*

**We apologize for the confusion. Indeed, the SSS in the CDW zone was also lower than the entire area in 2009. However, what we intended to address is that the stations at the CDW zone in 2010 where were not previously characterized by low SSS in 2009. To clarify this, this sentence has been slightly modified to become "*The higher CR in 2010 could be attributed to vigorous respiration of phytoplankton, especially at stations in the CDW zone that were not previously characterized by low sea surface salinity in 2009.*" in this revision. Please also refer to our response to your comment 10 for more details on this issue.**

*2. Keywords, line 51: It would be better to be more specific. Perhaps you could add "flooding", "CDW", "freshwater discharge" and also "Yangtze River"?*

**Thank you for the valuable suggestion. The suggested keywords have been added, with the exception of "CDW" since it is a non-standard abbreviation.**

*3. Comments on Figure 1: 1) The color of the SSS contour plot is confusing to read. Usually with salinity contours, the bluer the fresher and the redder the saltier, 2) you could add the color bars for Figures 1 and 2 both, and 3) please increase the font size.*

**Thank you for pointing out the common color code contours for salinity. We have now changed our figure to conform to norms in the field. In addition, the following changes have also been made: 1) color bars were added, 2) the font size was increased per your suggestion, and 3) the symbols for the sampling stations were enlarged. Collectively, we believe your comments have aided in the creation of an improved and easier to read figure. Please also refer to our reply to comment 10 of Reviewer #2 on a similar issue.**

*4. Comments on Table 1: 1) I think it is critical to compare if there are significant differences between the entire area and CDW zone for all variables in each year as well as overall all 2 years combined, and 2) why zooplankton values are not reported? They are in Table 2, so it should be able to calculate them.*

**Thank you for the valuable suggestion. We do agree that it is important to rule out whether there was a difference between the CDW zone and the other areas in the ECS in each year. However, we tended not to compare between**

the entire area and the CDW zone since they were dependent on each other. As you pointed out, significant differences were evident for almost all variables (Table 1) between the CDW zone and the other regions in the ECS in each year, except for nitrate and phosphate in 2009. To avoid confusion but still provide this information to the reader, we choose to portray this result in the table caption since the comparison on Table 1 was already complicated enough. However, we will be happy to show it in Table 1 if you still feel that is necessary.

As for zooplankton values, they were only collected for the whole water column, and not only for the surface water. Therefore, they were not presented in Table 1 in a comparable format.

**5.** *Comments on Table 2: 1) To be consistent with Table 1, you could use brackets for values in the CDW zone instead of parentheses. Also, in Table 1 you could report average values for the entire area (what are in parentheses for now) without parentheses and their ranges in parentheses instead (e.g. SSS: 32.62+/-2.07 (23.80-34.11)).*

**Good point! For consistency, parentheses have been replaced by brackets in Table 2, as suggested.**

**As for your suggestion on the format of Table 1, the average values for the entire ECS and the CDW zones were reported in both parentheses and brackets, as they were in Table 2. Therefore, we prefer to keep them in this format for consistency. Hopefully, that is understandable. But, we will be happy to modify it if you still think there is a better way to present these results, as we want them to be as easy to interpret as possible.**

**6.** *Results & discussion, line 226: Please clarify and write more clearly if 2003 was also an anomalous flooding year and how does its area of CDW zone compare with that in 2010.*

**Thank you for the valuable suggestion. To clarify, this sentence has been slightly modified to become "*A low transparency value was documented in June 2003 in the ECS, during which the CDW area was 43.1 x10$^3$ km$^2$ (~40% of the CDW area of the 2010 flood; Chen et al., 2009), and the average values for the ECS and the CDW were 70.9% and 66.0%, respectively (C.C. Chen, unpublished data).*"**

**7.** *Results & discussion, line 235: Add the sentence what the response time of phytoplankton bloom is to flooding events if reference exists.*

**We do agree with your suggestion, and it indeed makes much more sense to add a sentence on this topic. There is, however, little data on this issue. Based on our own dataset in coastal lagoons and limited literature in estuary systems, we have added the following sentence: "*In estuarine and coastal regions, phytoplankton blooms normally occur within 2-3 weeks after a heavy rainfall event (e.g., Hsieh et al., 2012; Meng et al., 2015; Mulholland et al., 2009).*"**

*8. Results & discussion, line 272: I think you need to be cautious about relying solely on N/P elemental ratio for discussing the status of nutrient limitation. The fact that during the 2010 flood chlorophyll was positively correlated with nitrate and silicate (lines 290) but not to (at alpha = 0.05) phosphate (line 291) suggests limitations of nitrate and silicate predominantly, and to a lesser extent, of phosphate on phytoplankton productivity. This is opposite to your statement that 2010 was more likely affected by phosphate limitation. Similarly, phytoplankton might be limited by all three nutrients in 2009 (lines 294-296). Other suggestion is that it would better to focus only on the area in which nitrate and phosphate values were below detection limits as indicated in Table 1.*

**Good point! Yes, you are correct in noting that the insignificant relationship between Chl *a* and phosphate concentration suggests that biomass was not limited significantly by phosphate levels in the 2010 flood. Interestingly, it appears that the insignificant linear relationship (*p* = 0.09) between Chl *a* and [phosphate] in the surface water in 2010 was likely driven by one station with a low Chl *a* value (1.1 mg Chl m$^{-3}$), but a high phosphate concentration (1.7 µM; Fig. 1f). The linear regression (Chl *a* vs. phosphate) became statistically significant if this potential outlier was excluded from the analysis (*p* < 0.001). Furthermore, the phosphate concentration in the surface water during the 2010 flood was low, with a mean±SD value of 0.17±0.30 µM, and it was similar to the value observed in 2009 (0.13±0.17 µM). In addition, the high N:P molar ratio in the CDW during the 2010 flood was found to be 40.4. Therefore, except for the high phosphate station, phytoplankton biomass in the 2010 flood was more likely limited by phosphate, in terms of both nutrient availability and the N:P molar ratio.**

**The areas where nitrate and phosphate were not present at detectable levels in 2010 were mostly outside of the CDW zone, especially for nitrate. Even though a significant linear relationship was observed between Chl *a* and nitrate concentration in 2010, the average nitrate value was still high (10.3 µM) in the CDW zone. Based on the significant relationships observed between Chl *a* and nitrate or silicate, this suggests that nitrate and silicate**

might be the limiting factors for phytoplankton growth during the 2010 flood. However, in addition to nitrate and silicate, phosphate might be the most important limiting factor for phytoplankton growth due to its low concentration level and high N:P molar ratio in 2010. In this manuscript, therefore, we try to emphasize that growth of phytoplankton was more likely limited by phosphate, especially in the CDW in 2010. To clarify, this sentence has been slightly modified to become "*Phytoplankton biomass might have also be limited by nitrate and silicate concentrations in 2010. Based on nutrient levels and the N/P molar ratio, however, phytoplankton growth was more likely limited by phosphate, especially in the CDW zone during the 2010 flood (please refer to Sect. 3.2 for other details.).*" Hopefully, this explanation and these data can satisfy your inquiry.

**9.** *Results & discussion, line 413-414: I do not follow. Please see below comments on Figures 3-5.*

We apologize for the confusion. In our previous version of the manuscript, we had simply tried to reduce the number of panels in the figure. Therefore, both data for 2009 and 2010 were combined into one panel in Fig. 3 for Chl *a* and bacteria. As for the results mentioned in the text, we might have jumped one step ahead to describe the results found in Fig. 5 (please refer to our response to your comment 10 for other details.). As you point out, Fig. 3 suggests that the CR might be dominated by phytoplankton and bacterioplankton in either 2009 or 2010. To clarify this, this and next sentence have been modified to "*These results suggest that phytoplankton and bacterioplankton were two important components that contributed substantially to the CR rate either in 2009 or 2010.*"

**10.** *Comments on Figures 3, 4, 5: When two variables are regressed onto each other, the slope of regression could be used to assess the strength of coupling between the two or control of one to another – the higher slope, the strong coupling/control, where even a small change in X-values results in a huge response in Y-values. In this regard, for Figure 3 I would consider stronger control of chlorophyll and bacterial biomass on CR in 2010 than in 2009 if you do inter-year comparisons. Figure 4 might also be similarly interpreted (the stronger control of PP on CR in 2010 than in 2009). The same goes for Figure 5. Additionally, if the unit of chlorophyll is converted to that of bacterial biomass, the relative strength of control by each on CR could also be inferred.*

We do agree that the slope of the regression line could be used to assess the

strength of coupling between two parameters. As in our reply to your previous comment regarding Fig. 3, we tried to analyze and understand whether or not the CR rates were related to phytoplankton and/or bacterioplankton biomass. Even though significant relationships were found between CR and 1) Chl *a* and 2) bacteria, this still could not explain why the CR was higher in 2010 than in 2009. Therefore, Fig. 5 were used to further explore whether differences in phytoplankton or bacterioplankton biomass contributed to the higher CR observed in 2010. Interestingly, the extent of the inter-annual (2010 minus 2009) difference in CR was significantly related to differences in Chl *a* concentration ($p < 0.001$) and bacterial biomass ($p < 0.01$; Fig. 5). Among the positive CR differences (i.e., 20 of 33 Sts.), 15 stations were also characterized by positive differences in Chl *a* concentrations, but only two stations had positive differences in bacterial biomass. Interestingly, the stations with positive Chl *a* concentration differences were mostly located within the CDW region in 2010. These results suggest that the higher CR during the 2010 flood might be attributed to phytoplankton, especially in the CDW. As for Fig. 4 (i.e., CR vs. PP), it has been removed from this revision since we only have PP for 2010 as your pointed out and suggested. Based on these analyses, therefore, we try to state in the Abstract that "*The higher CR documented in 2010 could be attributed to vigorous respiration of phytoplankton, especially at stations in the CDW zone that were not previously characterized by low sea surface salinity in 2009.*" For your reference, the modified Fig. 3 with Chl *a* converted to carbon units (C:Chl a of 52.9; Chang et al., 2003) is also presented as below:

[Figure]

Chang, J, F.-K. Shiah, G.-W. Gong, K.-P. Chiang. (2003). Cross-shelf variation in carbon-to-chlorophyll a ratios in the East China Sea, summer 1998. Deep-Sea Res. II, 50: 1237-1247.

**11.** *Results & discussion, line 468: This is only true relative to global average values in coastal oceans. I do not think you could say this unless you compare with P/R value in the non-flooding year, which you do not have the data for.*

**Yes, you are correct that we do not have P/R data for 2009. Therefore, this paragraph has been removed per your suggestion.**

**12.** *Results & discussion, lines 487-499: I suggest you provide a full summary of values calculated from endmember mixing equations, including equations and how to calculate uncertainties, RMSE, etc. Also, you might want to more emphasize that the calculations presented in this paragraph give you estimates that are expected by purely physical processes, without taking biological effect into account. Deviation from what is estimated based on endmember mixing model is due to biological effect, which you nicely described in the following paragraph.*

**The following new paragraph has been added to the revised manuscript to clearly explain the simulation of $fCO_2$ under a conservative mixing model, including equations and uncertainty analysis: "*Provided that the proportional contributions from freshwater and seawater endmembers are $f_1$ and $f_2$ ($f_1+f_2=1$), respectively, the conservative mixing TA and DIC values for a given water sample can be expressed by the following equations:***

$$TA_{mix}=TA_{fw}xf_1+TA_{sw}xf_2$$
$$DIC_{mix}=DIC_{fw}xf_1+DIC_{sw}xf_2$$

***where the subscripts "mix", "fw", "and "sw" represent values of conservative mixing, freshwater, and seawater endmembers, respectively. The TA and DIC data reported by Zhai et al. (2007) for Changjiang in summer were used as the freshwater endmember (both $TA_{fw}$ and $DIC_{fw}=1743$ $\mu$mol kg$^{-1}$), and the surface data at station K in July 2009 and 2010 were chosen to represent the seawater endmember ($TA_{sw}=2241$ $\mu$mol kg$^{-1}$ and $DIC_{sw}=1909$ $\mu$mol kg$^{-1}$ in 2009; $TA_{sw}=2240$ $\mu$mol kg$^{-1}$ and $DIC_{sw}=1904$ $\mu$mol kg$^{-1}$ in 2010). Subsequently, the hypothetical $fCO_2$ from conservative mixing was calculated from the $TA_{mix}$ and $DIC_{mix}$ data using the Excel macro CO2SYS version 2.1 (Pierrot et al. 2006), in which the carbonic acid dissociation constants were adopted from Mehrbach et al. (1973) and refitted by Dickson and Millero (1987). The uncertainty in this simulation mainly derives from the errors in the estimations of $TA_{mix}$ and $DIC_{mix}$. Assuming the errors of the calculated $TA_{mix}$ and $DIC_{mix}$ are $\pm5$ $\mu$mol kg$^{-1}$, this may result in an uncertainty of $\pm13$ $\mu$atm in the simulated $fCO_2$.***"**

**Technical Corrections:**

*1. Line 129: Replace "estimated" with "calculated"*

**Thank you for so thoroughly reviewing our manuscript and providing many valuable and constructive suggestions. It has been revised accordingly, and we hope you will find a superior manuscript.**

*2. Line 159: Replace "multiple" with "multiplied"*

**Thanks! The suggested change has been made.**

*3. Line 189: Avoid subjective words like "devastating".*

**Thanks! The suggested change has been made. We have used "great" in place of "devastating" in this revision.**

*4. Line 240: "immense" to "large"*

**Thanks! The suggested change has been made.**

*5. Line 264: Replace "16" with canonical Redfield ratio for N:P of 16 or so.*

**Thanks! The suggested change has been made.**

*6. Line 418: Awkward phrase "to gain greater insight".*

**This phrase has been replaced by "*In a further analysis*" in this revision.**

*7. Lines 433-444: Please be specific.*

**A short sentence has been added to point out that CR might be attributed to variable phytoplankton assemblages.**

*8. Line 463: I think it is "overestimated" given the vertical profile of P and R?*

**Yes, you are right. Thank you for pointing out this error!**

*9. Line 484-486: Please be specific and give numbers from the calculation.*

**The temperature effect on *f*$CO_2$ variation has been specifically explained in the revised sentence: "*The effect of temperature on the large variation in fCO₂ observed between the 2009 non-flooding period and the 2010 flood was trivial; the SST difference of 0.7ºC between 2009 and 2010 would only equate to a fCO₂ decrease of approximately 10 μatm (Table 1).*"**

**Referee #2**

**General comments:**

*Review of Chung-Chi Chen et al submitted to Biogeosciences. The aim of this paper is stated to 'reveal the effects of riverine input of dissolved inorganic nutrients on the plankton communities that support heterotrophic processes in the East China Sea shelf ecosystem between periods of non-flooding and flooding. Generally the topic of the paper is clearly introduced as a comparison of data collected during summer surveys of the ECS in July 2009 and 2010 with 2010 being a year when exceptional river flows from the Changjiang river impacted the coast waters of the ECS.*

*The methods are reasonably clearly described with references to several previous papers by the research team. However the collection of zooplankton needs more explanation – if they were vertical hauls through the water column give the depth range. Were the zooplankton preserved in formalin prior to counting?. . ..Also it is rather non-standard to use GF/F filters to collect 14C labelled phytoplankton following incubations. How significant was the loss of small phytoplankton ie <1um on the 14C uptake rates. Also as this 14C data was only collected during the 2010 survey I suggest it could be removed from the paper. Determining oxygen respiration rates from dark incubation of enclosed water samples by difference between initial fixed samples and final incubated bottles using the Winkler method to analyse for dissolved oxygen is a standard approach. However based on only two initial and two final replicates I suggest will yield low precision measurements. It is standard practise to use at least 4 replicates of initial and final bottle measurements. The precision stated is only really the difference divided by the mean of two replicates and I would suggest rather unreliable.*

We appreciate that you so thoroughly reviewed our manuscript, and you have provided many valuable and constructive suggestions. We also thank you for agreeing with our dataset's ability to be used to compare between flooding and non-flooding periods. We have taken your comments, as well as those of the other reviewers, very seriously in preparing this revised manuscript. Overall, we feel that the comments were very helpful and contributed to a greatly improved manuscript.

We also apologize for the unclear statements found within the "Materials and Methods" section, and we have done our best to clarify these in the revision. For example, the depth range of sampling and the preservation (with 10% buffered formalin) of zooplankton have now been mentioned. As for primary production, we do agree with your suggestion. Therefore, the related treatise of PP has been removed from the manuscript (including what was previously Fig. 4).

**Per your first major concern, it is a tedious and labor-intensive to make community respiration (CR) measurements *in situ*. We did include 672 and 692 CR measurements (initial + dark incubation) made in 2009 and 2010, respectively. Based on our previous measurements, our duplicate measurements have high precision (e.g., Chen et al. 2003). Therefore, duplicate, instead of triplicate, samples were incubated at each sampling depth. Hopefully, this is understandable.**

*My main problem with this paper however is the section labelled Results and Discussion. This section of the paper is 18 pages long! If the paper is to be resubmitted I strongly recommend that the results and discussion are presented as two different sections and the discussion section greatly shortened. The discussion and interpretation of the data currently included in the paper is at best speculative and in many places vague with the word 'might' used very frequently in numerous sentences. For example Page 19 lines 326-328 Page 19 line 340 Page 21 lines 323-375 Page 22 line 392 plus many more scattered throughout this section.*

**Thank you for your constructive suggestion. In the previous version, we indeed intended to discuss our results by analyzing them in concert with those of the primary literature. This was because that we did not want to rule out any potential causal factors for the observed outcomes in this study. Therefore, as you correctly pointed out, the manuscript was long and speculative; there are so many potential relationships that could have explained our data. We have tried to make it seem less speculative. In this revision, the ambiguous statements have been largely removed, as evidenced by a dramatic reduction in the page number (from 20 to 16 pages, even including of adding 2 pages which response to Reviewer #1). In addition and per your next comment, the Conclusions section has also been greatly shortened (from 3 to 2 pages) by removing less significant findings.**

*The conclusion section also needs to be much shorter and report the studies main findings without including too many references to other studies. In summary I strongly recommend this paper only be considered for publication if following resubmission the results and discussion are rewritten as separate sections and the discussion is greatly shortened and written less speculatively.*

**Wow! This is amongst the toughest comments we have ever received, and we struggled for a few days to respond. We understand that the long (20 pages) Results and Discussion was difficult to get through in the previous version of the manuscript. Although a combination of these two sections is permitted by**

*Biogeosciences*, we appreciate your concern and want to make this dataset and manuscript as accessible as possible. Rather than dividing the two sections, we have instead made an effort to shorten this combined section by removing the most speculative sections. Therefore, a significant amount of text has been removed, including: 1) almost all of the vaguely worded statements in the discussion and 2) text, tables, and figures related to primary production. This can be evident by a dramatic reduction in the page number (from 20 to 16 pages, even including of adding 2 pages which response to Reviewer #1) in this combined section as stated in our previous response to your comment. We also greatly shortened the conclusion per your suggestion (from 3 to 2 pages). Hopefully, you will now find that the Results and Discussion of this revision is less speculative than the previous version. If, however, you still find this section to be too long and continue to deem it necessary to split into a traditional Results section followed by a short Discussion section, we will be pleased to do so in the next incarnation of the manuscript.

**Specific Comments:**

*1. Page 2 line 42; 'vigorous plankton metabolic activities especially phytoplankton ' – rather vague- be more specific eg respiration? Production?*

We had somewhat deliberately worded it this way because we don't actually know why CR differed. To clarify, this sentence has been modified to "*The higher CR documented in 2010 could be attributed to vigorous respiration of phytoplankton, especially at stations in the CDW zone that were not previously characterized by low sea surface salinity in 2009.*" in this revision.

*2. Page 2 line 43 define 'SSS'*

Thanks! The "SSS" has now been replaced by "sea surface salinity" in this revision, though we were sure to define it upon its first usage in the text.

*3. Page 2 line 44 '. . .zooplankton might be . . .' far too vague in abstract.*

You are right, and we do agree with your comment. This sentence has been clarified to become "*…zooplankton were…*" in this revision.

*4. Page 5 line 72 line avoid using the word 'tremendous'*

The word "tremendous" has been replaced by "large" in the revised manuscript.

**5.** *Page 5 line 78 and elsewhere delete 'psu' salinity has no units now.*

**Thank you for pointing this out; "*psu*" has been deleted throughout this revision.**

**6.** *Page12 line 211 'previously documented values' – be more specific ie when?*

**We agree with your suggestion, and the actual comparison times (i.e., summer) has been specifically mentioned in this revision.**

**7.** *Page 13 line 230 change 'trailing' to 'previous'*

**Thank you! The suggested change has been made.**

**8.** *Page 15 line 261 the single high phosphate concentration also evident on figure 1 looks to be an analytical anomaly.*

**We have re-checked our logbook and did not find any calculation or instrument setting errors that could have resulted in this anomalous data point. Indeed, we seriously considered removing it, as doing so would facilitate our explanation of the data. We discuss this matter above, as the first reviewer also noted this strange result. For example, the linear regression (Chl *a* vs. phosphate) became statistically significant if this outlier was excluded from the analysis (*p* < 0.001; please also refer to our reply to comment 8 of reviewer #1 for more details). To avoid data tampering/mining, we have opted to keep this data point in this revision, even though it heavily influences the regression analysis.**

**9.** *Page 17 line 304 and table 2 data. I do not believe it is useful or that accurate to estimate the total chlorophyll a etc in the ECS. I suggest deleting table 2.*

**Reviewer #1 stated *"Nonetheless, the authors conducted a fairly good job of synthesizing what they have learned based on their current and others' previous observations and various indices and metrics (e.g. volumetric values in surface water, averaged over the depth of euphotic zone, and depth-integrated values for the entire ECS and the Changjiang Diluted Water (CDW) region)".* Indeed, we tried to provide as much oceanographic data as possible to elucidate how pelagic ecosystems respond to flooding. In Table 1, the average values of Chl *a* and bacterioplankton have been presented for 2009 and 2010. There was no effect of year on Chl *a* levels due to large variation in the dataset. As the region influenced by the river was much larger in 2010, it is difficult to know if the differences between mean values resulted from differences in ecosystem composition and function within this region or**

from the differences in the total area affected. Table 2 then provides another viewpoint to examine the effect of flooding on the response of the total biomass and other biological variables over the entire ECS and in the CDW zone. To provide the best estimation of total biomass, Surfer 11 (Golden Software, Inc.) was applied. In addition, zooplankton was one of the important contributors to CR during the flooding period, and the zooplankton data are only presented in Table 2 in this manuscript. For above reason, it seems necessary to keep Table 2 in this revision. However, to response to your concern, the chlorophyll data have been removed from Table 2. Hopefully, our explanation can satisfy your concerns.

*10. Page 46 and 47 Figure 1 and 2. The contour plots are not very clear. The sampling locations need to be more clearly indicated by lager clear symbols.*

Thank you for pointing out the lack of clarity in the contour plots; reviewer 1 also raised a similar issue. These figures have been modified to be easier to read. For instance, the color code for salinity was changed, color bars were added, the font size was increased, and the sampling station symbols were enlarged. Collectively, we now feel that's Figures 1-2 look more better, and we hope you do, as well.

*11. Figure 3 Although the relationships shown apparently are significant- the considerable scatter is not very convincing. If the one high chlorophyll point is removed from figure 3a is the relationship still significant? The relationships might be more usefully illustrated if the data from each year is shown on separate plots ie 2009 in upper figure and 2010 on lower figure with axis ranges the same on both figures.*

Even though we expected quite a degree of scatter in our data due to extensive temporal and spatial variation found in nature, we have still taken this suggestion as an opportunity to re-examine our dataset. The positive, linear relationship between CR and Chl *a* was still statistically significant even after the one high chlorophyll data point was removed from this analysis: $y = 31.5 x + 26.1$ ($r^2 = 0.36$; $p < 0.001$; $N = 167$).

We had plotted data from both surveys years together to reduce the number of panels, and, more importantly, to more readily allow for a visual comparison. For your reference, the new figure has been created as your suggestion as below. We will be glad to replace the original figure with this new one if you still deem it necessary.

---

## Author Response (AR2)

Dear Prof. Robinson:

We sincerely appreciate that you took such a great effort to process our manuscript. Even though it took a long time for this procedure, we do understand its difficulty, especially in finding potential reviewers. We now re-submit our revised manuscript entitled, "The influence of episodic flooding on a pelagic ecosystem in the East China Sea" [manuscript no: bg-2016-246], and you may find it from submission system of *Biogeosciences* (the tracked version has also been attached below where most of the changes are marked in red). We have substantially revised the manuscript in response to comments from reviewer's #3, and our detailed responses to the reviewer comments are listed below. In addition, we also thank the reviewer # 1 and 2 for agreeing with our previous revised manuscript. They do provide very constructive and valuable comments to improve our manuscript.

In this revision, we have taken the reviewer's comments very seriously in preparing this revised manuscript. In general, we are confident that we have been able to respond clearly and reasonably to these comments. Overall, we feel that the reviewer comments were very helpful and contributed to a greatly improved manuscript. We are confident that it is now suitable for publication in *Biogeosciences*.

We look forward to your decision concerning our manuscript.

With Best regards,

Chung-Chi

**Responses to reviewers' comments on ms no: bg-2016-246 "The influence of episodic flooding on pelagic ecosystem in the East China Sea" (Chen, Gong, Chou, Chung, Hsieh, Shiah, and Chiang)**

**Referee #3**

**General comments:**

*As noted by two previous referees, this manuscript provides a useful account of the distribution of several physico-chemical and biological variables in the East China Sea during contrasting situations of discharge of the Changjian (Yangtze) River. The authors have made a reasonable effort to address the suggestions of the referees. However, I have some additional comments that I would recommend to take into account before publication.*

*Microzooplankton is an important contributor to plankton community respiration, but this component of the food web was not measured in the study. In addition, the 330 µm mesh net used for zooplankton sampling (line 144) is likely to have failed to capture small components of the mesozooplankton. Curiously, the authors cite Calbet and Landry (2004) to support their sentence that "Zooplankton are the most important contributors to plankton CR" (line 307). However, Calbet and Landry (2004), as the title of their paper indicates, refer basically to microzooplankton, not to mesozooplankton, and state in their abstract that "The estimated contributions of microbial grazers to total community respiration are of the same magnitude as bacterial respiration". The authors refer to this problem in only two lines (378-379) at the end of the more than three pages of section 3.3 ("Effects of . . . flooding on plankton community respiration". I understand that at this point, it is impossible to obtain the missing microzooplankton data, but this drawback should be acknowledged from the beginning of the CR results and all the discussion should be framed accordingly, taking also into account that correlation is not causation. (e. g., sentences like these in lines 365-366 "phytoplankton and bacterioplankton might be the most important components contributing to CR", and many others should be reconsidered and improved).*

We appreciate that you so thoroughly reviewed our manuscript, and you have provided many valuable and constructive suggestions. We have taken your comments very seriously in preparing this revised manuscript. Overall, we feel that the comments were very helpful and contributed to a greatly improved manuscript.

Regarding the zooplankton, thank you for pointing out this ambiguous statement. To clarify, it has been slightly modified to become "*Zooplankton, especially microzooplankton, are amongst the most important contributors to plankton CR*" (line 314). We also added a sentence to remind the reader that microzooplankton were not measured and excluded from our analysis in this study. In addition, to avoid confusion, the size of measured zooplankton, i.e., > 330 $\mu$m, was also enclosed in brackets in the statement on this regard. Also, thank you for understanding that it is impossible to obtain the missing microzooplankton data in this study. To emphasize the potential impact of microzooplankton on CR or other plankton communities in the ESC, a few sentences on this regard have been added into section 3.2.

*The abstract states (lines 33-34) that the study had a focus on community respiration (CR). However, in the Results and Discussion section, this question appears only after 8-9 pages of dealing with other topics. I would recommend to explain more specifically what will be the topics dealt with in the manuscript (lines 84-93) and to modify the abstract accordingly. In this way, the reader will know what to expect when looking at the results.*

Thank you for the valuable suggestion. To state the objective more specifically, the sentence has been modified to become "***The main objective of this study was to reveal the effects of riverine input, particularly the associated DIN, on the plankton activities (e.g., phytoplankton, heterotrophic bacteria, and zooplankton (>330 $\mu$m)) and how they impact on CR in the ECS between periods of non-flooding and flooding***" (lines 89-91) at the end paragraph of the introduction. Also, it has also been slightly modified at the abstract. Hopefully, the change can clarify the objective of this study.

**Other comments:**

1. *Indicate somewhere in the Methods (rather than in line 360) that total plankton biomass is the "summed biomass of phytoplankton, bacterioplankton, and zooplankton".*

Good point! This sentence has been added in the Methods section (2.3 Biological variables) shown as "***To compare, total plankton biomass was the summed biomass of phytoplankton, bacterioplankton, and zooplankton over the $Z_E$***". (lines 133-135)

2. *Line 170. Explain what is the "threshold discharge rate".*

We apologize for the ambiguous statement. We tried to use "threshold discharge rate" to set the criteria for freshwater discharge rate that might cause flooding. To clarify, this sentence has been slightly modified to become "***the suggested discharge rate for flooding was 4-6 x 10$^4$ m$^3$ s$^{-1}$***" (line 172) in the revision.

3. *Lines 193-194. It should be "Regarding the Ze", I presume.*

Yes, you are right. Thank you for pointing out the typos. It has been corrected in this revision. (lines 195-196)

4. *Lines 196 and following. It would be better to use "transmittance" rather than "transparency".*

**Thank you for the valuable suggestion. This word has been replaced as suggested. (line 198)**

5. *Lines 237-239. Clarify if the N/P ratio of 22.3 +- 20.9 applies to all the "ECS". (this seems the case, since the CDW is mentioned next).*

**We apologize for the unclear statement. Yes, you are right that the N/P ratio of 22.3 was mean value of the entire ECS. To clarify, this sentence has been slightly modified to become "*during which the mean molar ratio of nitrate to phosphate (N/P) over the entire ECS was 22.3 ± 20.9*" (line 240) in this revision.**

6. *Lines 413-414. fCO₂ varied from 375.4 to 439.8 as SSS varied from 20.38 to 33.96 or in the opposite sense?*

**We like it. It significantly improved our statement. This sentence has then been modified to become "*it varied from 375.4 to 439.8 $\mu$atm as salinity varied from 20.38 to 33.96*" (lines 423-424) since the positive trend has been found between $f$CO₂ and SSS.**

[revised manuscript text omitted]

$y = - 45.62 - 0.62\ x$
$(r^2 = 0.31; p = 0.001; N = 31)$

Fig. 7